# Exploring the Boundary of Diffusion-based Methods for Solving Constrained Optimization

## Abstract

Diffusion models have achieved remarkable success in generative tasks such as image and video synthesis, and in control domains like robotics, owing to their strong representation capabilities and proficiency in fitting complex multimodal distributions. However, their full potential in solving Continuous Constrained Optimization problems remains largely underexplored. Our work commences by investigating a two-dimensional constrained quadratic optimization problem as an illustrative example to explore the inherent challenges and issues when applying diffusion models to such optimization tasks and providing theoretical analyses for these observations. To address the identified gaps and harness diffusion models for Continuous Constrained Optimization, we build upon this analysis to propose a novel diffusion-based framework for optimization problems called **DiOpt**. This framework operates in two distinct phases: an initial warm-start phase, implemented via supervised learning, followed by a bootstrapping training phase. This dual-phase architecture is designed to iteratively refine solutions, thereby improving the objective function while rigorously satisfying problem constraints. Finally, multiple candidate solutions are sampled, and the optimal one is selected through a screening process. We present extensive experiments detailing the training dynamics of DiOpt, its performance across a diverse set of Continuous Constrained Optimization problems, and an analysis of the impact of DiOpt's various hyperparameters. The official implementation of DiOpt is provided in `https://anonymous.4open.science/r/diopt-iclr-DFFE`.

## 1 Introduction

Constrained optimization with hard constraints constitutes a cornerstone of real-world decision-making systems, spanning critical applications from power grid operations (Pan et al., 2020; Ding et al., 2024b; Cain et al., 2012; Shi et al., 2017) and wireless communications (Du et al., 2024) to robotic motion planning (Li et al., 2025a; Chi et al., 2023). Traditional numerical methods (Nocedal & Wright, 1999; Zimmerman et al., 2011; Wächter & Biegler, 2006) face a fundamental trade-off: either simplify problems through restrictive relaxations (e.g., linear programming approximations) or endure prohibitive computational costs, both unsuitable for safety-critical and real-time systems. Learning-based approaches (Donti et al., 2021; Park & Van Hentenryck, 2023; Zamzam & Baker, 2020; Zhang & Zhang, 2022; Jiang et al., 2024; Zhao & Barati, 2024) emerged as promising alternatives by training neural networks to predict solutions directly, yet they suffer from two critical limitations, as shown in Figure 1: 1) Deterministic one-to-one generation methods exhibit poor performance in handling multi-value mappings (Liang & Chen, 2024). 2) The learned constraint points show limited overlap with the feasible region.

Recent efforts to address these limitations have turned to diffusion models (Ho et al., 2020; Song et al., 2020a;b), leveraging their multimodal sampling capacity to generate diverse solution candidates (Li et al., 2025a; Pan et al., 2024). While this mitigates the single-point failure risk, state-of-the-art methods still struggle with systematic constraint satisfaction due to persistent distributional misalignment. In addition, they rely heavily on supervised training with labeled datasets, a practical bottleneck given the NP-hard nature of many constrained optimization problems. Additionally, these diffusion-based optimization methods (Li et al., 2025a; Pan et al., 2024) often require complex guidance or projection procedures to refine solutions during sampling, resulting in slow inference speeds that scale poorly with problem dimensionality.

Besides, some diffusion-based methods attempt to generate constraint-satisfying points through gauge mapping (Li et al., 2025b). However, these methods rely on a large number of feasible samples for training, despite the fact that the optimal solution in typical optimization problems is a single point in the distribution.

To address the absence of an efficient way that bridges the diffusion model and constrained optimization. We first analyze the problem of directly applying the supervised diffusion training paradigm in optimization. Then, to address these problems, we present DiOpt, a self-supervised diffusion framework that fundamentally rethinks how generative models interact with constrained solution spaces. Concretely, DiOpt introduces a target distribution designed to maximize overlap with the constrained solution manifold and develops a boot-

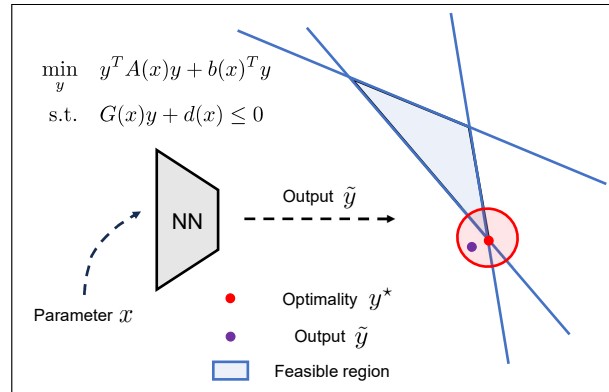

Figure 1: A schematic geometric interpretation of feasibility challenges in learning-based optimization. The feasible region (blue) and output distribution of neural network (red), e.g., a Gaussian output by a diffusion model, fundamentally exhibit a small overlap, particularly under high dimensionality and multiple constraints. This distributional misalignm ent breaks constraint satisfaction.

strapped self-training mechanism that assigns weights to candidate solutions based on the severity of constraint violations and optimality gaps.

Due to the instability of sampling, the current round of diffusion model sampling may not necessarily outperform its previous round, and directly using these points for training could degrade the model. Thus, DiOpt also introduces a look-up table to store historically optimal sampling points, thereby ensuring effective training. Our contributions are threefold:

- We reveal an inherent limitation of all existing diffusion-based methods for optimization, which leads to low feasibility in the generated solutions. This limitation arises from the fact that existing diffusion-based methods all follow a supervised paradigm, which leads to a mismatch between the target distribution of the diffusion solver and the feasible region of the problem to be solved, especially in high-dimensional spaces.

- To handle this inherent limitation, we propose DiOpt as a remedy for all existing diffusion-based methods. By introducing an extra weighted bootstrapping mechanism after supervised learning, the feasibility of the diffusion solver is enhanced while maintaining decent optimality. Notably, DiOpt is the first method that resolves the mismatch limitation and introduces the bootstrapping training mechanism into learning-based optimization methods.

- We evaluate the proposed DiOpt method in a diverse range of nonconvex optimization problems, including synthetic concave QP problems, power grid control, motion retargeting, and wireless power allocation. This comprehensive evaluation encompasses various optimization scenarios with convex and nonconvex objectives and constraints, demonstrating the method's generalizability across different cases.

## 2 RELATED WORKS

**Learning to Optimize.** To address the high computational cost of classical optimization solvers, Learning to Optimize (L2O) has emerged as a promising approach that leverages machine learning techniques to solve real-world constrained optimization problems. The L2O methods can be generally categorized into two groups: 1) assisting traditional solvers with machine learning techniques to improve their efficiency or performance; 2) approximating the input-output mapping of optimization problems using data-driven models. In the first category, reinforcement learning (RL) has been widely adopted to design better optimization policies for both continuous (Li & Malik, 2016) and discrete decision variables (Liu et al., 2022; Tang et al., 2020). Additionally, neural networks have been used to predict warm-start points for optimization solvers, significantly reducing convergence time (Baker, 2019; Dong et al., 2020). In the second category, deep learning models have been

employed to directly approximate solutions for specific problems. For instance, Fioretto et al. (2020); Chatzos et al. (2020) utilize neural networks to solve the OPF problems efficiently. To further improve constraint satisfaction, recent works have integrated advanced techniques into the training process. For example, Donti et al. (2021) introduce gradient-based correction, while Park & Van Hentenryck (2023) incorporate primal-dual optimization methods to ensure the feasibility of the learned solutions.

**Neural Solvers with Hard Constraints.** Despite the challenge of devising general-purpose neural solvers for arbitrary hard constraints, there are also some tailored neural networks (with special layers) for constrained optimization, especially for combinatorial optimization. In these methods, the problem-solving can be efficiently conducted by a single forward pass inference. For instance, in graph matching, or more broadly the quadratic assignment problem, there are a series of works (Wang et al., 2019; Fey et al., 2020) introducing the Sinkhorn layer into the network to enforce the matching constraint. Another example is the cardinality-constrained problem; similar techniques can be devised to ensure the constraints (Brukhim & Globerson, 2018; Wang et al., 2023; Cao & Li, 2024). However, these layers are specifically designed and cannot be used in general settings as addressed in this paper. Moreover, it often requires ground truth for supervision, which cannot be obtained easily in real-world cases.

**Generative Models for Constrained Optimization.**

Generative methods, characterized primarily by sampling from noise, involve models that transform random noise (typically a standard Gaussian distribution) into a specified distribution. In the past few years, a considerable number of studies with diverse methodologies have focused on this area. One category of methods is derived from modifications to the sampling process of classical diffusion models (Zhang et al., 2024; Kurtz & Burdick, 2024; Pan et al., 2024; Kong et al., 2024). By directly transforming the optimization problem into a probability function to replace the original score function in the sampling process, this class of methods has been developed without training procedure or with training required only for certain unknown objective constraints. However, the absence of training typically results in a high inference cost, and these methods are often limited to problems with soft constraints, that is, constraints are just required to satisfy a certain probability. Another category employs neural networks to process noise for transformation into a specified distribution, such as methods based on CVAE (Li et al., 2023) or GAN (Salmona et al., 2022). Furthermore, there are methods based on diffusion models: Briden et al. (2025) simulates the distribution of optimization problems through compositional operations on multiple score functions; Li et al. (2025a) forces the model to learn feasible solutions by adding violation penalties; (Liang & Chen, 2024) provides a theoretical guarantee for such methods. This process can be applied multiple times to improve the quality of the solution. To enforce the feasibility of generated solutions, PDM (Christopher et al., 2024) performs a projection after each diffusion step, while CGD (Kondo et al., 2024) proposes a targeted post-processing method for the problems it addresses. For combinatorial optimization, T2T (Li et al., 2024a) and Fast T2T (Li et al., 2024b) also propose a training-to-testing framework.

Table 1: Comparison among existing L2O algorithms across various dimensions.

| Method | Target Task | Real-time Requirement | Low Data Demand | Solution Diversity | Learning Paradigm | Non-differentiable Objective |
|---|---|---|---|---|---|---|
| DC3 (Donti et al., 2021) | continuous, hard | ✓ | ✓ | ✗ | self-supervised | ✗ |
| DiffOPT (Kong et al., 2024) | continuous, soft | ✓ | ✓ | ✓ | supervised | ✓ |
| MBD (Pan et al., 2024) | continuous, soft | ✗ | ✓ | ✓ | N/A | ✓ |
| DiffuSolve (Li et al., 2025a) | continuous, hard | ✓ | ✓ | ✓ | supervised | ✗ |
| RectFlow (Liang & Chen, 2024) | continuous, hard | ✓ | ✗ | ✓ | supervised | ✓ |
| T2T (Li et al., 2024a) | combinatorial, hard | ✗ | ✓ | ✓ | supervised | ✗ |
| DiOpt(*) | continuous, hard | ✓ | ✓ | ✓ | mixed | ✓ |

***Remark.*** However, most of the above methods are trained in a supervised learning paradigm, which tends to suffer from the small overlapping problem mentioned in Figure 1. In contrast, the proposed DiOpt trained in a bootstrapping paradigm can converge to the mapping to the near-optimal feasible region that has a large overlap with the feasible region and does not introduce extra cost on supervised data collection. Besides, it is worth noting that DiOpt focuses on constrained nonconvex optimization with continuous (real-valued) variables with smooth or nonconvex objectives and constraints (e.g., robotics, smart grids, communication systems), which fundamentally differs from combinatorial optimization (Li et al., 2024a;b; Karalias & Loukas, 2020; Sanokowski et al., 2024; 2025) (e.g., TSP) in both problem structure and methodology. As shown in Table 1, we summarized the differences

between our method and several representative existing L2O approaches in terms of target task, data demand, and learning paradigm, etc.

## 3    PRELIMINARIES

**Learning for Hard-constrained Optimization.** Learning-to-optimize attempts to learn neural networks to generate solutions for a family of optimization problems as

$$
\begin{aligned}
\min_{\mathbf{y}} \quad & f(\mathbf{y}; \mathbf{x}) \\
\text{subject to} \quad & g_i(\mathbf{y}; \mathbf{x}) \leq 0 \quad i = 1, \cdots, m \\
& h_j(\mathbf{y}; \mathbf{x}) = 0 \quad j = 1, \cdots, n
\end{aligned}
\tag{1}
$$

where $\mathbf{y} \in \mathbb{R}^{d_y}$ is the decision variable of the optimization problem parameterized by $\mathbf{x} \in \mathbb{R}^{d_x}$. We can use machine learning techniques to learn the mapping from $x$ to its corresponding solution $\mathbf{y}^\star$ in an optimization problem family with a similar problem structure. With this mapping, the solution can be calculated faster and more efficiently compared with the classical optimization solver. Moreover, considering the differences between hard equality and inequality constraints, learning-to-optimize methods typically employ different mechanisms to ensure their satisfaction (Donti et al., 2021; Ding et al., 2024b).

**Diffusion Models.** Denoising diffusion probabilistic models (DDPM) (Ho et al., 2020) are generative models that create high-quality data by learning to reverse a gradual forward noising process applied to the training data. Given a dataset $\{\mathbf{y}_0^i\}_{i=1}^D$ for $\mathbf{y}_0^i \sim q(\mathbf{y}_0)$, the forward process $\{\mathbf{y}_{0:T}\}$ adds Gaussian noise to the data with pre-defined schedule $\{\beta_{1:T}\}$:

$$
q(\mathbf{y}_t \mid \mathbf{y}_{t-1}) := \mathcal{N}(\mathbf{y}_t; \sqrt{1 - \beta_t}\mathbf{y}_{t-1}, \beta_t \mathbf{I}).
\tag{2}
$$

Using the Markov chain property, we can obtain the analytic marginal distribution of conditioned on $x_0$:

$$
q(\mathbf{y}_t \mid \mathbf{y}_0) = \mathcal{N}(\mathbf{y}_t; \sqrt{\bar{\alpha}_t}\mathbf{y}_0, (1 - \bar{\alpha}_t)\mathbf{I}), \forall t \in \{1, \ldots, T\},
\tag{3}
$$

where $\alpha_t = 1 - \beta_t$ and $\bar{\alpha}_t = \prod_{s=0}^T \alpha_s$. Given $x_0$, it's easy to obtain a noisy sample by the reparameterization trick.

$$
\mathbf{y}_t = \sqrt{\bar{\alpha}_t}\mathbf{y}_0 + \sqrt{1 - \bar{\alpha}_t}\epsilon, \epsilon \in \mathcal{N}(\mathbf{0}, \boldsymbol{I}).
\tag{4}
$$

DDPMs use parameterized models $p_\theta(\mathbf{y}_{t-1} \mid \mathbf{y}_t) = \mathcal{N}(\mathbf{y}_{t-1}; \mu_\theta(\mathbf{y}_t, t), \Sigma_\theta(\mathbf{y}_t, t))$ to fit $q(\mathbf{y}_t \mid \mathbf{y}_{t-1}, \mathbf{y}_0)$ to reverse the forward diffusion process, where $\theta$ denotes the learnable parameters. The practical implementation involves directly predicting the Gaussian noise $\boldsymbol{\epsilon}$ using a neural network $\boldsymbol{\epsilon}_\theta(\mathbf{y}_t, t)$ to minimize the evidence lower bound loss. With $\boldsymbol{\epsilon}_t \sim \mathcal{N}(0, \mathbf{I})$, the loss in DDPM takes the form of:

$$
\mathbb{E}_{t \sim [1,T], \mathbf{y}_0, \boldsymbol{\epsilon}_t} \left[ ||\boldsymbol{\epsilon}_t - \boldsymbol{\epsilon}_\theta(\sqrt{\bar{\alpha}_t}\mathbf{y}_0 + \sqrt{1 - \bar{\alpha}_t}\boldsymbol{\epsilon}_t, t)||^2 \right].
\tag{5}
$$

After training, DDPM generates samples according to

$$
\boldsymbol{y}_{t-1} = \frac{1}{\sqrt{\alpha_t}} \left( \boldsymbol{y}_t - \frac{1 - \alpha_t}{\sqrt{1 - \bar{\alpha}_t}}\epsilon_\theta(\boldsymbol{y}_t, t) + \eta \cdot \sigma_t \boldsymbol{z} \right), \quad \boldsymbol{z} \sim \mathcal{N}(\mathbf{0}, \mathbf{I}), \quad t = T, ..., 1
\tag{6}
$$

where $\eta$ denotes the noise level.

## 4    WHY SUPERVISED DIFFUSION TENDS TO GENERATE INFEASIBLE SOLUTIONS

As mentioned in Section 1, how to apply diffusion models to constrained continuous optimization problems effectively and then generate near-optimal solutions without violating the complex (nonconvex) constraints remains largely unexplored. To investigate the underlying reason, we conduct a toy example, which applies diffusion models trained in a supervised paradigm to a two-dimensional QP problem with linear constraints. Figure 2 shows that this kind of diffusion model tends to generate points around the optimal solution, and the distribution is an approximate Gaussian distribution. Nevertheless, the feasible region only has a small overlap with the Gaussian ball, which leads to the diffusion solver violating the constraints with a high probability (i.e., the number of blue points (feasible) is greater than that of the green points).

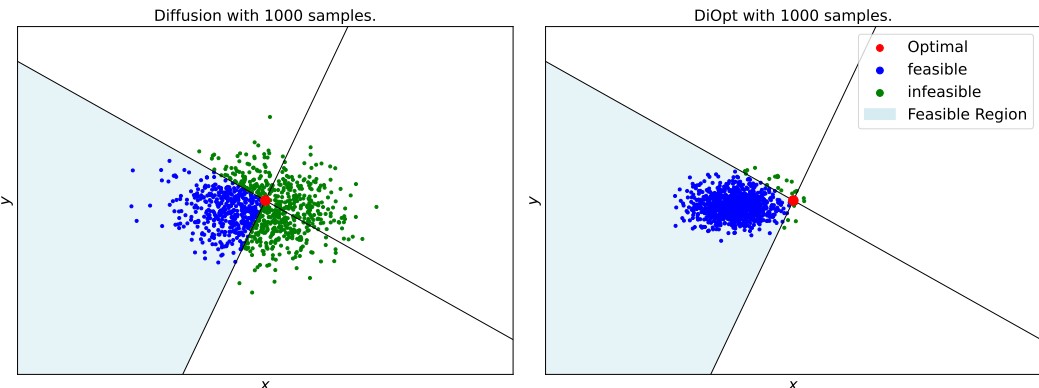

(a) Diffusion model trained in a supervised manner.   (b) Diffusion solver trained by DiOpt.

Figure 2: **Comparison between supervised diffusion and DiOpt on a toy example.** It can be observed that the distribution of diffusion trained in a supervised manner (a) approximates a Gaussian distribution centered around the optimal point, which leads to a low feasibility rate. For detailed settings of this toy example, please refer to Appendix F.2 for more details.

Based on this finding, we also present a theoretical proof, as Theorem 1, that in high-dimensional optimization problems, exemplified by linear programming, diffusion models trained via supervised learning become increasingly prone to constraint violations, with the violation probability growing exponentially towards one as the dimensionality increases.

**Theorem 1.** *(Feasibility in Linear Programming.)* *Given a d-dimensional linear programming problem of the form:*

$$\min_{y \in \mathbb{R}^{d_y}} c^T y \quad \text{subject to} \quad a_i^T y \leq b_i, \quad i = 1, \cdots, N,$$

*where $a_i$ represents a unit normal vector, drawn independently and uniformly from the unit sphere $S^{d_y - 1}$. Let $y^\star$ be the unique solution to this linear programming problem, and we define a neighborhood ball $B_\epsilon(y^\star) = \{y : \|y - y^\star\| \leq \epsilon\}$. For sufficiently small $\epsilon > 0$, there is an asymptotic bound on the probability that a point uniformly sampled from $B_\epsilon(y^\star)$ lies in the feasible region $\mathcal{C}$.*

$$\mathbb{P}_{x \sim B_\epsilon(y^\star)} (y \in \mathcal{C}) \approx \frac{1}{2^{d_y}}.$$

For more general nonlinear inequality constraints, one can similarly linearize them within an infinitesimal neighborhood and reduce to the case described in this theorem. This theorem also confirms our findings in the toy example in Figure 2. The proof leverages results from stochastic geometry (Cover & Efron, 1967; WENDEL, 1962), with details provided in Appendix A. Besides, we also provide a theoretical analysis to illustrate this issue cannot be resolved via multiple sampling in Appendix B. Hence, it suggests that training diffusion models with optimal samples is not an ideal choice for constrained optimization problems, and a more effective method needs to be developed to better incorporate the diffusion model into constrained optimization.

## 5 METHODOLOGY

To address the issue mentioned in Section 4, we propose DiOpt, a self-supervised **Di**ffusion-based learning framework for Constrained **Opt**imization in this section. As shown in Figure 3, DiOpt trains the diffusion model in a bootstrapping mechanism via weighted variational loss of diffusion and applies the candidate *solution selection* technique during the evaluation stage to further boost the solution quality. The training procedure of DiOpt is presented in Algorithm 1.

**Target Distribution for Diffusion Training.** To apply the diffusion model to optimization problems, we train it using the following objective:

$$\mathbb{E}_{\boldsymbol{x}, \boldsymbol{y}^\star, \epsilon, t} \left[ \|\epsilon - \epsilon_\theta(\boldsymbol{y}_t, \boldsymbol{x}, t)\|_2^2 \right] \tag{7}$$

where $\boldsymbol{y}^\star \in \mathbb{R}^{d_y}$ denotes the optimal solution corresponding to $\boldsymbol{x} \in \mathbb{R}^{d_x}$ and $\epsilon_\theta$ is the noise network of diffusion model $\mathcal{D}_\theta$. The near-optimal distribution learned by the diffusion model trained in this

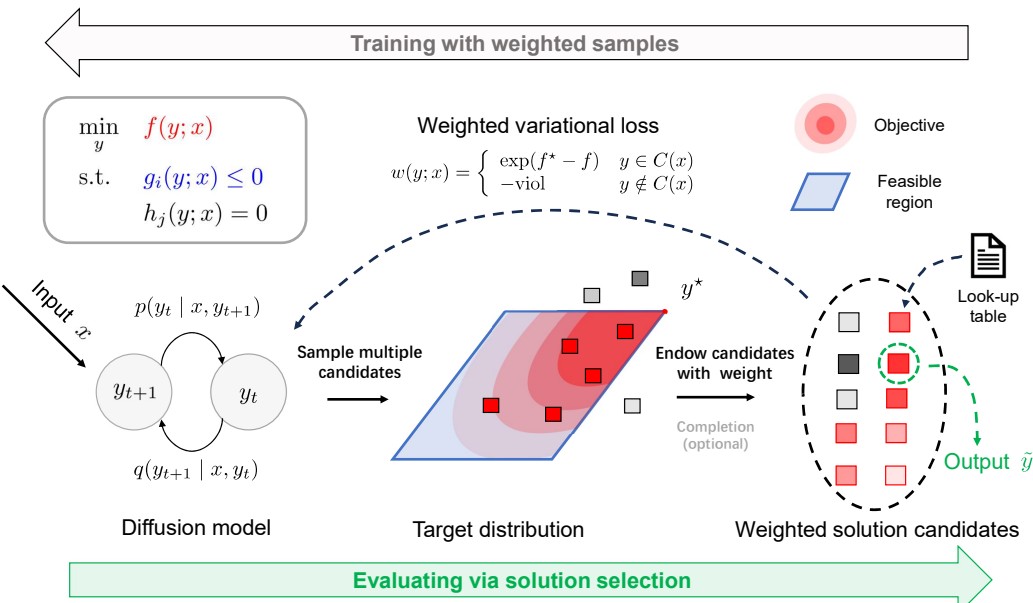

Figure 3: Training and evaluating procedure of DiOpt.

manner is ultimately similar to that shown in Figure 1. However, in high-dimensional settings, the intersection between the near-optimal region of the diffusion model and the feasible region becomes too small, which leads to infeasibility issues when applying the diffusion model to optimization problems. In that case, it is necessary to define another target distribution for diffusion models to enforce their constraint satisfaction. Referring to (Liang & Chen, 2024), we define the target distribution that corresponds to the near-optimal feasible region as

$$p(\boldsymbol{y}; \boldsymbol{x}) \sim \mathbb{I}_{\mathcal{C}(\boldsymbol{x})}(\boldsymbol{y}) \exp\left(-\beta f(\boldsymbol{y}; \boldsymbol{x})\right), \tag{8}$$

where $\boldsymbol{y} \sim \mathcal{D}_\theta(\boldsymbol{x})$, $\mathcal{C}(x)$ indicates the constraint region that satisfies $g_i(\boldsymbol{y}; \boldsymbol{x}) \leq 0, h_j(\boldsymbol{y}; \boldsymbol{x}) = 0$ and $\mathbb{I}_{\mathcal{C}(\boldsymbol{x})}$ is the indicator function that judges the satisfaction of the constraint.

**Training Diffusion with Bootstrapping.** While Liang & Chen (2024) has argued that it is essential to approximate the above target distribution rather than merely an optimal point, it is difficult to sample from (8) efficiently. Therefore, constructing a dataset based on the target distribution for training diffusion models is also impossible. To avoid this problem, we divert our attention to self-supervised learning. Motivated by (Ding et al., 2024a), we design a diffusion-based learning framework for constrained optimization, which implements diffusion training in a bootstrapping manner. To better adapt the weighted training mechanism to constrained optimization, we design entirely different weighting functions to address both constraint satisfaction and objective optimality. In addition, we introduce techniques such as a reset operation and a look-up table for higher feasibility and faster convergence. In this way, DiOpt can naturally converge to the target distribution without manufacturing a corresponding dataset.

Concretely, we try to utilize the parallelism of the diffusion model and generate a certain number of candidate points for one specific problem, and then endow the candidate points with weights related to the constraint violation and objective value. The diffusion model will be trained with these weighted candidate points according to:

$$\mathcal{L}(\theta) := \mathbb{E}_{\boldsymbol{x}, \boldsymbol{y}, \boldsymbol{\epsilon}, t}\left[\omega(\boldsymbol{y}; \boldsymbol{x}) \|\boldsymbol{\epsilon} - \boldsymbol{\epsilon}_\theta\left(\boldsymbol{y}_t, \boldsymbol{x}, t\right)\|_2^2\right], \tag{9}$$

Here, the weight can be viewed as the importance of training points. Hence, the diffusion model can approximately converge to the distribution defined by the weight function after enough iterations. Thus, the weight function design is essential to ensure that the diffusion model to converge to the target distribution. Based on that, we classify points into two cases and design two different weight

functions for them:

$$\omega(\boldsymbol{y}; \boldsymbol{x}) := \begin{cases} \exp\left(f^\star(\boldsymbol{x}) - f(\boldsymbol{y}; \boldsymbol{x})\right) & y \in \mathcal{C}(\boldsymbol{x}) \\ -\sum_i \max(g_i(\boldsymbol{y}; \boldsymbol{x}), 0) & y \notin \mathcal{C}(\boldsymbol{x}) \end{cases}, \tag{10}$$

where $f^\star(\boldsymbol{x})$ indicates the optimal objective value of the optimization problem with parameter $\boldsymbol{x}$. One of the principal ideas for this weight function is that all the feasible points have positive weights and the infeasible points have negative weights. In that case, the diffusion model will converge to the feasible region and then consider the optimality of the points inside the constraint region. Besides, it is worth noting that $f^\star(\boldsymbol{x})$ can be replaced with the estimated lower bound of the objective function. This term actually avoids the numerical explosion of the exponential function. Furthermore, to ensure that the trained model prioritizes feasibility over optimality, we reset the weight function every training epoch as follows:

$$\omega(\mathbf{y}; \mathbf{x}) = -\sum_i \max(g_i(\mathbf{y}; \mathbf{x}), 0) \tag{11}$$

***Remark.*** If we continue updating diffusion with the weight of (10), the output of the diffusion solver will resemble that of supervised learning and will be prone to generating infeasible points. The detailed proof can be referred to in the Appendix C. Hence, this formulation penalizes constraint violations ($g_i(\mathbf{y}; \mathbf{x}) > 0$), systematically steering the model toward the feasible region.

However, there is still a problem to be resolved in our weight function. As illustrated in (Ding et al., 2024a), the weight in (9) must always be positive. Hence, we perform a modification on the final weight when there exists a candidate point with a negative weight.

$$\widetilde{\omega}(\boldsymbol{y}; \boldsymbol{x}) = \max\left(\omega(\boldsymbol{y}; \boldsymbol{x}) - \bar{\omega}, 0\right), \quad \text{where } \bar{\omega} = \frac{1}{K_t} \sum_{i=1}^{K_t} \omega(\boldsymbol{y}_i; \boldsymbol{x}), \quad \boldsymbol{y}_i \overset{i.i.d}{\sim} \mathcal{D}_\theta(\boldsymbol{x}) \tag{12}$$

As shown in (Ding et al., 2024a), $\widetilde{\omega}$ is equivalent to $\omega$ for diffusion training, we can ensure the diffusion model converges to the target distribution with the modified weight $\widetilde{\omega}$. This leads to the following approach:

$$\mathbb{E}_{\boldsymbol{x}, \widetilde{\boldsymbol{y}}, \epsilon, t}\left[\widetilde{\omega}(\widetilde{\boldsymbol{y}}, \boldsymbol{x}) \|\epsilon - \epsilon_\theta(\widetilde{\boldsymbol{y}}_t, \boldsymbol{x}, t)\|_2^2\right], \quad \widetilde{\boldsymbol{y}} = \underset{\boldsymbol{y} \in \{\boldsymbol{y}_1, \ldots, \boldsymbol{y}_{K_t}\}}{\arg\max} \omega(\boldsymbol{y}; \boldsymbol{x}) \tag{13}$$

Due to the curse of dimensionality (Bellman et al., 1957), training the model purely based on bootstrapping may make it difficult for the diffusion model to explore sufficiently good solutions. Therefore, at the initial stage of training, we utilize (7) to train the diffusion model and guide it towards the vicinity of the optimal point. Afterwards, (13) is employed to further train the model, ensuring feasibility. Throughout all training epochs, the proportion of supervised learning is denoted as $r_s$.

**Handling of Equality Constraint.** We apply a standard technique to handling hard equality constraints in learn-to-optimize literature (Donti et al., 2021; Ding et al., 2024b). Specifically, equality constraints inherently reduce the degrees of freedom of the problem itself. For problems that have $n$ variables $m$ equality constraints $h_i(\boldsymbol{y}; \boldsymbol{x}) = 0$, we choose to divide the problem variables into two parts: $\boldsymbol{y} = \{\boldsymbol{y}_b, \boldsymbol{y}_n\}$, where $\boldsymbol{y}_b \in \mathbb{R}^{n-m}$ is the basic variables that denote the actual freedom degree of the problem, $\boldsymbol{y}_n \in \mathbb{R}^m$ is the nonbasic variables that denote the left variables.

Once the basic variables are given, the left nonbasic variables are the solution of equations $h_i(\boldsymbol{y}_b, \boldsymbol{y}_n; \boldsymbol{x}) = 0$. Therefore, we can utilize an equation solver to solve the equations and obtain the nonbasic variables. In that case, the equality constraints can be satisfied exactly. DiOpt also follows this paradigm to deal with the hard equality constraints. Concretely, diffusion solver just outputs the basic variables and uses an equation solver for the left part in DiOpt for problems with equality constraints.

Moreover, as mentioned in Section 3, for existing methods, handling hard inequality constraints while simultaneously maintaining equality constraints often incurs a substantial computational cost, as many gradient updates are required. Besides, it remains difficult to reliably ensure the satisfaction of the inequality constraints in practice. In that case, DiOpt is proposed to achieve better feasibility of hard inequality constraints.

**Quality Assurance Mechanism.** Due to the inherent randomness of diffusion model, we cannot guarantee that each round of sampling yields a better solution than the previous one. To ensure

---

**Algorithm 1** Bootstraping-based training process of DiOpt

---

**Input:** training dataset $X$, the objective function $f(\boldsymbol{y}; \boldsymbol{x})$ and constraints $h(\boldsymbol{y}; \boldsymbol{x}), g(\boldsymbol{y}; \boldsymbol{x})$, the noise network $\epsilon_\theta$ of diffusion model $\mathcal{D}_\theta$, look-up table $\mathcal{B}$, Number of Training Epochs $N_e$, Supervised Ratio $r_s$, Number of Training Samples $K_t$.

**for** $n = 0$ **to** $N_e - 1$ **do**
  **if** $n \leq \lfloor r_s \cdot N_e \rfloor$ **then**
    Train $\mathcal{D}_\theta$ by (7)
    Continue
  **end if**
  **if** t **mod** 2 = 0 **then**
    Reset the weight function as (10)
  **else**
    Reset the weight function as (11)
    // reset diffusion with feasible points
  **end if**
  **for** $\boldsymbol{x}_i$ **in** $X$ **do**
    $\boldsymbol{y}_1, \ldots, \boldsymbol{y}_{K_t} \overset{i.i.d}{\sim} \mathcal{D}_\theta(\boldsymbol{x}_i)$
    $\boldsymbol{y}_{\text{best}} \leftarrow \mathcal{B}(\boldsymbol{x}_i)$
    Endow $\boldsymbol{y}_1, \ldots, \boldsymbol{y}_{K_t}, \boldsymbol{y}_{\text{best}}$ with weights according to (12)
    Train $\mathcal{D}_\theta$ by (13)
    Update $\mathcal{B}(\boldsymbol{x}_i)$ with $\boldsymbol{y}_1, \ldots, \boldsymbol{y}_{K_t}$ by (14)
  **end for**
**end for**
**Return** $\mathcal{D}_\theta$

---

training stability, we maintain a look-up table $\mathcal{B} : \mathbb{R}^{d_x} \to \mathbb{R}^{d_y}$ that stores the best solution $y$ found so far for each input. Let $\boldsymbol{y}_{\text{best}}$ denote the optimal solution stored in $\mathcal{B}$ for a given problem parameter $\boldsymbol{x}$. The update rule for $\mathcal{B}$ is formulated as:

$$\mathcal{B}(\mathbf{x}) = \underset{y \in \{\boldsymbol{y}_1, \ldots, \boldsymbol{y}_{K_t}, \boldsymbol{y}_{\text{best}}\}}{\arg\max} \omega(\boldsymbol{y}; \boldsymbol{x}) \tag{14}$$

After training, we employ *Solution Selection* for inference:

$$\widetilde{\mathbf{y}} = \underset{\mathbf{y} \in \{\mathbf{y}_1, \ldots, \mathbf{y}_{K_e}\}}{\arg\max} \omega(\mathbf{y}; \mathbf{x}), \quad \mathbf{y}_1, \ldots, \mathbf{y}_{K_e} \overset{i.i.d}{\sim} \mathcal{D}_\theta(\mathbf{x}) \tag{15}$$

By selecting a sufficiently large $K_e$, we ensure a high probability of obtaining a near-optimal solution.

## 6 EXPERIMENT

Building upon the illustrative Toy Example demonstration in Figure 2b, we now conduct a comprehensive empirical evaluation of DiOpt on a range of more complex and challenging optimization tasks. These tasks include manually constructed problems (QP, QPSR, and CQP) as well as challenging real-world benchmarks such as AC Optimal Power Flow (ACOPF), a non-convex optimization problem in power systems, and Motion Retargeting (He et al., 2024), which maps human motion to a humanoid robot under kinematic constraints. CQP is a specially designed benchmark. Because of its distinctive objective function, it is more prone to infeasibility compared with other tasks. Spanning diverse domains, these problems highlight DiOpt's versatility and representation capability. Detailed experimental settings, ablation results, benchmark formulations and baseline settings are provided in Appendix D, E, F and Appendix G.

### 6.1 BENCHMARKING RESULTS: PERFORMANCE COMPARISON

**Table Notes.** In this section, we use IPOPT (Wächter & Biegler, 2006), a nonlinear optimization solver, as solver benchmark. Bold numbers indicate the best metric (marked with light blue in Table 2). "$N/A$" denotes an invalid or abnormal result. Objective marked with $\times$ indicate that the objective value is not meaningful due to infeasibility (Feasibility$< 50\%$). Conversely, an objective marked with $\checkmark$ indicates that the objective value is optimal with feasibility (Feasibility$\geq 50\%$).

Table 2: Performance comparison (mean ± std. dev.) across different methods. We takes $K_e = 64$ for DiOpt and correction steps 200 for DC3. The row highlighted in light blue indicates a "solver" (IPOPT here), which here is used only as a reference. All of the learning-based methods (DC3, DiOpt, Diffusion) were trained on the training set. All baselines were evaluated and reported on the test set, which is distinct from the training set.

| Problem | Method | Feasibility(%)↑ | Gap(%)↓ | Objective↓ | Time(s)↓ | Ineq Mean↓ | Ineq Max↓ | Ineq Num Viol↓ |
|---|---|---|---|---|---|---|---|---|
| QP | IPOPT | $100.00\% \pm 0.00$ | $0.00\% \pm 0.00$ | $-10.90 \pm 0.45$✓ | $0.63 \pm 0.11$ | $0.00 \pm 0.00$ | $0.00 \pm 0.00$ | $0.00 \pm 0.00$ |
| | DiOpt | $\mathbf{79.11\% \pm 0.00}$ | $\mathbf{3.45\% \pm 1.23}$ | $-10.53 \pm 0.47$✓ | $0.01 \pm 0.00$ | $0.00 \pm 0.00$ | $\mathbf{0.02 \pm 0.06}$ | $\mathbf{0.28 \pm 0.50}$ |
| | Diffusion | $0.00\% \pm 0.00$ | $0.97\% \pm 3.22$ | $-10.81 \pm 0.55$× | $0.01 \pm 0.00$ | $0.01 \pm 0.03$ | $0.58 \pm 0.65$ | $14.02 \pm 4.20$ |
| | DC3 | $22.93\% \pm 0.00$ | $34.98\% \pm 6.64$ | $-7.10 \pm 0.83$× | $0.01 \pm 0.00$ | $0.00 \pm 0.01$ | $0.48 \pm 0.64$ | $2.09 \pm 2.34$ |
| | MBD | $0.04\% \pm 0.00$ | $3588.43\% \pm 3743.72$ | $380.26 \pm 409.20$× | $0.01 \pm 0.00$ | $10.77 \pm 7.47$ | $81.30 \pm 50.46$ | $95.14 \pm 19.30$ |
| QPSR | IPOPT | $100.00\% \pm 0.00$ | $0.00\% \pm 0.00$ | $-9.77 \pm 0.42$✓ | $0.62 \pm 0.10$ | $0.00 \pm 0.00$ | $0.00 \pm 0.00$ | $0.00 \pm 0.00$ |
| | DiOpt | $\mathbf{81.87\% \pm 0.00}$ | $\mathbf{2.48\% \pm 0.78}$ | $-9.53 \pm 0.43$✓ | $0.01 \pm 0.00$ | $0.00 \pm 0.00$ | $\mathbf{0.01 \pm 0.04}$ | $\mathbf{0.23 \pm 0.47}$ |
| | Diffusion | $0.00\% \pm 0.00$ | $0.43\% \pm 2.87$ | $-9.73 \pm 0.51$× | $0.01 \pm 0.00$ | $0.01 \pm 0.02$ | $0.34 \pm 0.48$ | $11.84 \pm 2.82$ |
| | DC3 | $20.65\% \pm 0.00$ | $33.61\% \pm 7.00$ | $-6.49 \pm 0.78$× | $0.01 \pm 0.00$ | $0.00 \pm 0.01$ | $0.48 \pm 0.64$ | $2.03 \pm 2.18$ |
| | MBD | $0.04\% \pm 0.00$ | $4101.15\% \pm 4377.28$ | $390.88 \pm 428.73$× | $0.01 \pm 0.00$ | $11.00 \pm 7.84$ | $83.11 \pm 53.43$ | $95.84 \pm 19.66$ |
| CQP | IPOPT | $100.00\% \pm 0.00$ | $0.00\% \pm 0.89$ | $-37.18 \pm 0.70$✓ | $3.22 \pm 1.02$ | $0.00 \pm 0.00$ | $0.00 \pm 0.00$ | $0.00 \pm 0.00$ |
| | DiOpt | $\mathbf{69.95\% \pm 0.00}$ | $\mathbf{7.04\% \pm 1.36}$ | $-34.57 \pm 0.64$✓ | $0.01 \pm 0.00$ | $0.00 \pm 0.00$ | $\mathbf{0.05 \pm 0.22}$ | $\mathbf{0.75 \pm 1.82}$ |
| | Diffusion | $0.00\% \pm 0.00$ | $1.47\% \pm 0.85$ | $-36.68 \pm 0.71$× | $0.01 \pm 0.00$ | $0.00 \pm 0.00$ | $0.33 \pm 0.18$ | $11.27 \pm 1.90$ |
| | DC3 | $0.00\% \pm 0.00$ | $N/A$ | $N/A$× | $0.01 \pm 0.00$ | $N/A$ | $N/A$ | $N/A$ |
| | MBD | $0.00\% \pm 0.00$ | $45672.14\% \pm 23099.09$ | $-17005.74 \pm 8566.45$× | $0.01 \pm 0.00$ | $23.51 \pm 8.42$ | $602.93 \pm 214.30$ | $70.64 \pm 5.15$ |
| RETARGETING | IPOPT | $100.00\% \pm 0.00$ | $0.05\% \pm 1.22$ | $1.73 \pm 0.51$✓ | $4.36 \pm 21.500$ | $0.00 \pm 0.00$ | $0.00 \pm 0.00$ | $0.00 \pm 0.00$ |
| | DiOpt | $\mathbf{100.00\% \pm 0.00}$ | $\mathbf{0.65\% \pm 1.19}$ | $1.74 \pm 0.51$✓ | $0.01 \pm 0.00$ | $0.00 \pm 0.00$ | $\mathbf{0.00 \pm 0.00}$ | $\mathbf{0.00 \pm 0.00}$ |
| | Diffusion | $100.00\% \pm 0.00$ | $1.24\% \pm 2.52$ | $1.74 \pm 0.50$✓ | $0.01 \pm 0.00$ | $0.00 \pm 0.00$ | $0.00 \pm 0.00$ | $0.00 \pm 0.00$ |
| | DC3 | $95.86\% \pm 0.00$ | $30.16\% \pm 37.68$ | $2.14 \pm 0.53$× | $0.00 \pm 0.00$ | $0.00 \pm 0.00$ | $0.01 \pm 0.06$ | $0.04 \pm 0.20$ |
| | MBD | $0.00\% \pm 0.00$ | $169.20\% \pm 57.09$ | $4.49 \pm 1.08$× | $0.00 \pm 0.00$ | $0.06 \pm 0.01$ | $2.30 \pm 0.33$ | $1.00 \pm 0.00$ |
| ACOPF57 | IPOPT | $100.00\% \pm 0.00$ | $0.00\% \pm 0.00$ | $3.81 \pm 0.64$✓ | $0.42 \pm 0.04$ | $0.00 \pm 0.00$ | $0.00 \pm 0.00$ | $0.00 \pm 0.00$ |
| | DiOpt | $93.33\% \pm 0.00$ | $0.24\% \pm 0.91$ | $3.81 \pm 0.64$✓ | $0.02 \pm 0.00$ | $0.00 \pm 0.00$ | $0.00 \pm 0.01$ | $0.09 \pm 0.33$ |
| | Diffusion | $81.33\% \pm 0.00$ | $\mathbf{0.19\% \pm 0.71}$ | $3.81 \pm 0.64$✓ | $0.02 \pm 0.00$ | $0.00 \pm 0.00$ | $0.01 \pm 0.01$ | $0.26 \pm 0.56$ |
| | DC3 | $\mathbf{94.00\% \pm 0.00}$ | $0.40\% \pm 0.85$ | $3.82 \pm 0.63$× | $0.01 \pm 0.00$ | $0.00 \pm 0.00$ | $\mathbf{0.00 \pm 0.00}$ | $0.07 \pm 0.29$ |
| | MBD | $0.00\% \pm 0.00$ | $22.36\% \pm 14.01$ | $4.74 \pm 1.31$× | $1.28 \pm 0.00$ | $0.02 \pm 0.01$ | $1.30 \pm 0.54$ | $4.30 \pm 1.28$ |
| ACOPF118 | IPOPT | $100.00\% \pm 0.00$ | $0.00\% \pm 0.00$ | $13.11 \pm 1.22$✓ | $0.97 \pm 0.042$ | $0.00 \pm 0.00$ | $0.00 \pm 0.00$ | $0.00 \pm 0.00$ |
| | DiOpt | $\mathbf{84.33\% \pm 0.00}$ | $\mathbf{2.26\% \pm 0.11}$ | $13.41 \pm 1.23$✓ | $0.07 \pm 0.00$ | $0.00 \pm 0.00$ | $\mathbf{0.01 \pm 0.01}$ | $\mathbf{0.20 \pm 0.40}$ |
| | Diffusion | $0.00\% \pm 0.00$ | $2.90\% \pm 0.21$ | $13.49 \pm 1.25$× | $0.07 \pm 0.00$ | $0.00 \pm 0.00$ | $0.38 \pm 0.14$ | $9.10 \pm 1.30$ |
| | DC3 | $43.00\% \pm 0.00$ | $2.49\% \pm 0.19$ | $13.44 \pm 1.24$× | $0.06 \pm 0.00$ | $0.00 \pm 0.00$ | $0.02 \pm 0.03$ | $0.73 \pm 0.76$ |
| | MBD | $0.00\% \pm 0.00$ | $37.39\% \pm 12.81$ | $17.86 \pm 0.33$× | $5.34 \pm 0.00$ | $0.03 \pm 0.01$ | $4.50 \pm 2.41$ | $23.16 \pm 1.33$ |

We define an inequality constraint $g_i(\boldsymbol{y}; \boldsymbol{x})$ as violated if $g_i(\boldsymbol{y}; \boldsymbol{x}) > \epsilon$, where $\epsilon$ is the threshold for inequality violation. The following metrics quantify inequality constraint violations: "Ineq Mean" (mean violation value), "Ineq Max" (maximum violation value), and "Ineq Num Viol" (number of violated constraints). The threshold $\epsilon$ is fixed at $0.01$ for all experiments.

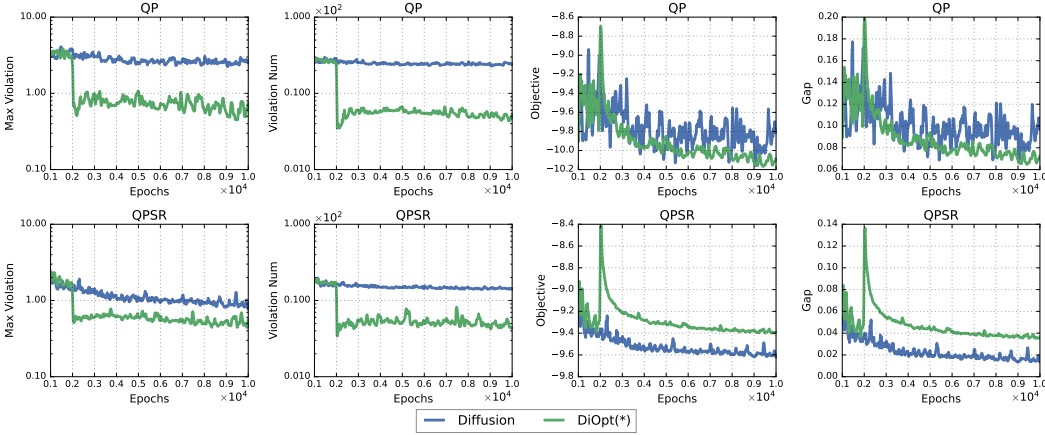

Figure 4: Training progression across different optimization tasks, with each row representing a unique optimization problem (indicated in subtitles) and each column showing specific performance metrics. The horizontal axes track training epochs, while the vertical axes display corresponding metric values with detailed scales labeled for each plot. Notably, **DiOpt achieves sudden gains in constraint satisfaction approximately at 2000 epochs when compared with Diffusion**. The bootstrapping mechanism initially causes an upward perturbation in the objective function and optimality gap, which then progressively decays to near-optimal.

**Results.** Table 2 summarizes the performance of all methods across the benchmark tasks. DiOpt consistently attains the smallest optimality gap and exhibits the highest feasibility on the majority of tasks, with the exception of ACOPF57. These results indicate that DiOpt effectively exploits the multi-solution characteristics of diffusion models and benefits from the proposed bootstrapping strategy, achieving a favorable balance between feasibility and near-optimality.

In contrast, MBD shows irregular behaviors that can be attributed to its equality-completion procedure, which introduces unstable point selection during inference. However, without equality-completion, it is impossible to satisfy equality constraints with pure sampling. The DC3 method fails on the CQP task due to numerical instability: as certain variables diverge to large magnitudes, the CQP loss function decreases without bound, ultimately driving the training objective toward negative infinity and preventing convergence.

Diffusion-based approaches generally achieve smaller optimality gaps because the learned Gaussian distributions concentrate around near-optimal regions. However, as analyzed in Section 4, these methods suffer from severe feasibility degradation. Except for a few tasks, their feasibility remains close to zero even when evaluating with $K_e = 64$. Detailed task configurations and additional analyses are provided in Appendix F.

## 6.2 ABLATION STUDY ON RESET OPERATION

In this section, we conduct a simple ablation study to illustrate the effect of the reset mechanism introduced in Section 5. Without this mechanism, once DiOpt discovers a point that is extremely close to the optimum, the subsequent weighted selection tends to repeatedly choose this single point. This behavior can cause DiOpt to degenerate toward supervised diffusion to some extent. The reset step prevents such collapse by steering the method toward the feasible region rather than a specific solution, thereby preserving feasibility. We evaluate this effect on the QPSR task, and the results are reported in Table 3. As shown, incorporating the reset mechanism leads to substantially improved feasibility. A more detailed theoretical explanation, along with additional experiments on broader benchmarks, is provided in Appendix C.

| Use Reset | Feasibility(%)↑ |
|:---------:|:---------------:|
| False     | 70.03           |
| True      | **81.19**       |

Table 3: Effects of the reset mechanism on QPSR task. Enabling reset in training improves feasibility by preventing collapse to a single near-optimal point.

## 6.3 TRAIN PROCEDURE

This section illustrates the training dynamics of DiOpt and Diffusion as observed on the QP and QPSR tasks. For training dynamics results across other tasks, please refer to Appendix E.5. As shown in Figure 4, once bootstrapping begins (**2000** epochs), DiOpt immediately exhibits a significant reduction in constraint violation metrics.

Meanwhile, the objective-related metrics initially experience a slight increase, followed by a gradual decrease toward near-optimal values. This behavior indicates that, after bootstrapping is activated, DiOpt first guides the sampling distribution into the interior of the feasible region, and subsequently steers it progressively toward the neighborhood of the optimal solution. In this way, DiOpt achieves feasibility improvement while preserving near-optimality.

## 7 CONCLUSION AND OUTLOOK

This work first explored diffusion models for constrained optimization, revealing that purely supervised methods struggle with feasibility as constraint dimensionality increases. To address this, we proposed DiOpt, a diffusion training framework combining supervised and self-supervised learning. DiOpt guides sampling towards feasibility using a target distribution and improves optimality via weighted self-supervised training. Extensive experiments validate DiOpt's effectiveness, demonstrating improved feasibility and preserved objective quality compared to baselines. Despite its strengths, DiOpt faces limitations in training efficiency due to its self-supervised nature and the costly feasibility completion needed for equality constraints.

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

## A    PROOF FOR THEOREM 1

**Theorem 1 (*Feasibility in Linear Programming.*)**  *Given a d-dimensional linear programming problem of the form:*

$$\min_{y \in \mathbb{R}^{d_y}} c^T y \quad subject \ to \quad a_i^T y \leq b_i, \quad i = 1, \cdots, N,$$

*where $a_i$ represents a unit normal vector, drawn independently and uniformly from the unit sphere $S^{d_y-1}$. Let $y^\star$ be the unique solution to this linear programming problem, and we define a neighborhood ball $B_\epsilon(y^\star) = \{y : \|y - y^\star\| \leq \epsilon\}$. For sufficiently small $\epsilon > 0$, there is an asymptotic bound on the probability that a point uniformly sampled from $B_\epsilon(y^\star)$ lies in the feasible region $\mathcal{C}$.*

$$\mathbb{P}_{y \sim B_\epsilon(y^\star)} (y \in \mathcal{C}) \approx \frac{1}{2^d}.$$

*Proof.* Motivated by (Cover & Efron, 1967), we can find that, for each linear equation $a_i^T y = 0$, it defines a hyperplane that divides $\mathbb{R}^d$ into two half-spaces, i.e.,

$$A_i := \{y : \text{sgn}(a_i^T y) = 1\}, -A_i := \{y : \text{sgn}(a_i^T y) = -1\}. \tag{16}$$

In that case, we can observe that $d_y$ linearly independent linear equations will partition the n-dimensional space $\mathbb{R}^{d_y}$ into $2^{d_y}$ distinct regions, intersecting at the origin. In other words, let $\delta_i = \text{sgn}(a_i^T y)$ and then we have $2^{d_y}$ choices for $\{\delta_1, \cdots, \delta_{d_y}\} = \{\pm 1, \cdots, \pm 1\}$. Considering the equivalence of each region, a random point sampled uniformly from the unit ball in the n-dimensional space falls into any particular region with an expected probability of $\frac{1}{2^{d_y}}$.

Back to the linear programming (LP) problem (16), it follows from the Fundamental Theorem of Linear Programming (Nocedal & Wright, 1999), that if an LP problem has a unique optimal solution $y^\star \in \mathbb{R}^{d_y}$, the optimal solution must lie at the vertex, which is the intersection of $d$ linearly independent inequality constraints.

By translating the coordinate system with the optimal solution $y^\star$ as the origin, we can find that computing the probability $\mathbb{P}_{y \sim B_\epsilon(y^\star)} (y \in \mathcal{C})$ is actually equivalent to the problem described above. In that case, we have

$$\mathbb{E}_{y \sim B_\epsilon(y^\star)} [\mathbb{P} (y \in \mathcal{C})] = \frac{1}{2^{d_y}}.$$

$\square$

## B    MULTIPLE SAMPLING CANNOT RESCUE SUPERVISED DIFFUSION

It has been analysed that we can improve the solution quality via multiple sampling of diffusion in (Liang & Chen, 2024). However, we need to clarify here that the solution quality cannot be efficiently enhanced with the supervised diffusion training method due to the mismatch between the desired distribution

$$p_d \propto \begin{cases} \exp(-\|y - y^\star\|^2), & y \in \mathcal{C} \\ 0, & otherwise \end{cases}$$

and the actual diffusion target distribution $p_{target} \propto \exp(-\|y - y^\star\|^2)$, as we mentioned in Section 4.

**Lemma 1.  (*Feasibility under Multiple Sampling.*)**  *For a supervised diffusion model, even with multiple samples from itself, it is still very hard to obtain a feasible sample, and the probability that all $m$ samples are located outside the constraint region is bounded by*

$$\Pr_{y_1, \cdots, y_m}(\arg\max_{y_1, \cdots, y_m} \omega(y; x) < 0 \,| y_1, \cdots, y_m \sim \mathcal{D}_\theta) \leq \left(1 - \frac{1}{2^{d_y}} + C_1 e_\theta^{1/4} + C_2 T^{-1/2}\right)^m.$$

*Proof.* According to Theorem 1 in (Liang & Chen, 2024), with $m$ samples from the supervised diffusion model, we have

$$
\begin{aligned}
\Pr(y \sim \mathcal{D}_\theta \in \mathcal{C}) &= \Pr(y \sim \mathcal{D}_\theta \in \mathcal{C}) - \Pr(y \sim \text{dataset} \in \mathcal{C}) + \Pr(y \sim \text{dataset} \in \mathcal{C}) \\
&\geq \Pr(y \sim \text{dataset} \in \mathcal{C}) - |\Pr(y \sim \mathcal{D}_\theta \in \mathcal{C}) - \Pr(y \sim \text{dataset} \in \mathcal{C})| \\
&\geq \Pr(y \sim \text{dataset} \in \mathcal{C}) - \sup_A \{|\Pr(A;\theta) - \Pr(A;\text{dataset})|\} \\
&= \Pr(y \sim \text{dataset} \in \mathcal{C}) - \text{TV}(p_\theta(y;x), p_{target}(y;x)) \\
&= \Pr(y \sim \text{dataset} \in \mathcal{C}) - \Pr(y \sim \mathcal{D} \in \mathcal{C}) + \Pr(y \sim \mathcal{D} \in \mathcal{C}) \\
&\qquad\qquad\qquad\qquad\qquad - \text{TV}(p_\theta(y;x), p_{target}(y;x)) \\
&\geq \Pr(y \sim \mathcal{D} \in \mathcal{C}) - \text{TV}(p_{target}(y;x), p_d(y;x)) - \text{TV}(p_\theta(y;x), p_{target}(y;x))
\end{aligned}
$$

where $\Pr(y \sim \mathcal{D}_\theta \in \mathcal{C})$ denotes the probability of sampling one feasible solution from supervised diffusion, $\Pr(y \sim \text{dataset} \in \mathcal{C})$ denotes the probability of sampling one feasible solution from the dataset distribution, and $\Pr(y \sim \mathcal{D} \in \mathcal{C})$ denotes the probability of sampling one feasible solution from the desired distribution. $p_\theta(y;x)), p_d(y;x)$, and $p_{target}(y;x)$ represent the probability density of the supervised diffusion model, desired distribution and dataset distribution given condition $x$, respectively. Then, according to (Liang & Chen, 2024), here we can split the total variation distance between the actual diffusion model distribution $p_\theta^{discrete}(y;x)$ and the dataset distribution $p_{taeget}(y;x)$ into three parts:

$$
\text{TV}\left(p_\theta^{discrete}(y;x), p_{target}(y;x)\right) \leq \underbrace{\text{TV}\left(p_{taeget}(y;x); p_\theta(y;x)\right)}_{\text{learning error}} + \underbrace{\text{TV}\left(p_\theta(y;x); p_\theta^{discrete}(y;x)\right)}_{\text{discretization error}}
$$

$$
\leq C_1 e_\theta^{1/4} + C_2 T^{-1/2}
$$

where $C_1, C_2$ are positive constant, $e_\theta$ is the generalization error of noise network and $T$ is the number of diffusion step. The detailed definition can be referred to in Appendix A of (Liang & Chen, 2024).

Besides, applying the Theorem 1, the total variation between $p_{target}$ and $p_d$ can be approximated as

$$
\begin{aligned}
\text{TV}(p_{target}(y;x); p_d(y;x)) &= \int_{y \notin \mathcal{C}} |(p_{target}(y;x) - p_d(y;x)| \, dy \\
&= \int_{y \notin \mathcal{C}} |(p_{target}(y;x)| \, dy \\
&\approx 1 - \frac{1}{2^{d_y}}
\end{aligned}
$$

Finally, we can achieve the probability that there exists no feasible solution under $m$ times sampling from the supervised diffusion

$$
\Pr\left(\underset{y_1,\cdots,y_m \sim \mathcal{D}_\theta}{\arg\max} \, \omega(y;x) < 0\right) \leq \left(1 - \Pr(y \sim \text{dataset} \in \mathcal{C}) + 1 - \frac{1}{2^{d_y}} + C_1 e_\theta^{1/4} + C_2 T^{-1/2}\right)^m.
$$

Considering all $y$ in the dataset are the optimal solution, we can simplify this formula as

$$
\Pr\left(\underset{y_1,\cdots,y_m \sim \mathcal{D}_\theta}{\arg\max} \, \omega(y;x) < 0\right) \leq \left(1 - \frac{1}{2^{d_y}} + C_1 e_\theta^{1/4} + C_2 T^{-1/2}\right)^m.
$$

For a high-dimensional problem, $\left(1 - \frac{1}{2^{d_y}} + C_1 e_\theta^{1/4} + C_2 T^{-1/2}\right)^m$ is obviously very close to 1 even with a sufficiently large number of samples from diffusion. That is why multiple sampling cannot rescue the infeasibility of supervised diffusion.

$\square$

## C   ANALYSIS FOR FUNCTION OF RESET OPERATION

As we mentioned in Section 5, if we continue updating diffusion with the weight of (10), the output of the diffusion solver will resemble that of supervised learning and will be prone to generating

infeasible points. For simplicity, we consider the case that the diffusion solver has been well trained and all the points generated by it are feasible in one optimization problem parameterized by $\boldsymbol{x}$. Then, the weight function will actually be

$$\omega(\boldsymbol{y}) = \exp\left(f^{\star}(\boldsymbol{x}) - f(\boldsymbol{y}; \boldsymbol{x})\right). \tag{17}$$

Let the initial distribution of generated points from the well-trained diffusion solver be $\rho_0(\boldsymbol{y}; \boldsymbol{x})$. Then, after $N$ iterations of weighted bootstrapping using (17), the distribution of generated points will be

$$\begin{aligned}
\rho_N(\boldsymbol{y}; \boldsymbol{x}) &\propto \rho_0(\boldsymbol{y}; \boldsymbol{x}) \prod_{i=1}^{N} \exp\left(\beta\left(f^{\star}(\boldsymbol{x}) - f(\boldsymbol{y}; \boldsymbol{x})\right)\right) \\
&= \rho_0(\boldsymbol{y}; \boldsymbol{x}) \exp\left(\beta N \cdot \left(f^{\star}(\boldsymbol{x}) - f(\boldsymbol{y}; \boldsymbol{x})\right)\right),
\end{aligned} \tag{18}$$

according to the reweighting technique, where $\beta > 0$ is a small value determined by the learning rate. Hence, when $N \to \infty$, $\rho_N(\boldsymbol{y}; \boldsymbol{x})$ will converge to the Dirac distribution $\delta(x - x^{\star})$. This is equivalent to the supervised diffusion solver trained with optimal points. In contrast, we can avoid this problem by redistributing the probability density across the feasible region with $\rho_1 \propto \rho_0 \exp(\beta\left(f^{\star}(\boldsymbol{x}) - f(\boldsymbol{y}; \boldsymbol{x})\right)) \approx \rho_0$ using the reset operation.

In addition, we provide an experiment in Table 4 to illustrate the impact of Reset. As shown in the table below, not applying Reset leads to varying degrees of feasibility degradation in DiOpt. The underlying reason is exactly as explained in (18).

| Problem | Reset | Feasibility(%)↑ |
|---------|-------|-----------------|
| QP | False | $75.39\% \pm 0.00$ |
| | True | $\mathbf{86.63\%} \pm \mathbf{0.00}$ |
| QPSR | False | $70.03\% \pm 0.00$ |
| | True | $\mathbf{81.19\%} \pm \mathbf{0.00}$ |
| CQP | False | $83.79\% \pm 0.00$ |
| | True | $\mathbf{87.07\%} \pm \mathbf{0.00}$ |

Table 4: Effects of Reset on QP, QPSR and CQP Tasks. On all three tasks, applying Reset achieves better feasibility.

## D  HYPERPARAMETER SETTINGS

In this paper, the following hyperparameters are involved:

| Parameter | Symbol | Value (Main Experiments) |
|-----------|--------|--------------------------|
| Supervised learning ratio | $r_s$ | 0.2 |
| Number of evaluation points | $K_e$ | 64 |
| Number of training samples | $K_t$ | 16 |
| Number of training epochs | $N_e$ | 10000 |
| Number of diffusion steps | $T$ | 100 |
| Noise schedule coefficient | $\eta$ | 1 |

Table 5: Experimental hyperparameters used in the main text.

The number of sampling points refers to the number of samples generated by the model $\mathcal{D}_\theta$ on the validation and test sets. The number of training samples, $K_t$, denotes the number of samples used during training. Note that $K_t \neq K_e$, as a large training sample size would significantly increase training time. The supervised learning ratio $r_s$ determines the portion of training steps that use supervised learning. Specifically, DiOpt is trained with optimal solutions during the first $\lfloor N_e \cdot r_s \rfloor$ steps. After that, the buffer $\mathcal{B}$ is initialized, and training continues in the same manner. By setting $r_s = 1$, the procedure reduces to standard Diffusion as described in the paper. The number of diffusion steps corresponds to the variable $T$ in (6).

Our noise network $\epsilon_\theta$ consists of a time encoding module and a backbone network. The time encoding module maps each timestep into a 32-dimensional vector using sinusoidal positional embeddings, followed by a two-layer MLP with 512 hidden units and Mish activation. The backbone network comprises four linear layers, each with 512 hidden units. The input dimension is $d_y + d_x + 32$. Each layer is followed by a Mish activation function. The final output of diffusion model has dimension $d_y$. In all experiments, we apply equality constraint completion for all baselines, but do not perform inequality correction for diffusion-based methods.

The experiments were performed on a high-performance workstation featuring an Intel Core i9-14900K processor (24 cores, 32 threads, 6.0 GHz turbo frequency) paired with dual NVIDIA GeForce RTX 4090D GPUs (24GB GDDR6X VRAM each) for accelerated deep learning computations. The system was equipped with 128GB DDR5 RAM. The Ubuntu 22.04.5 LTS operating system with Linux kernel 6..8.0-79-generic and NVIDIA driver 550.163.01 provided an optimized environment for GPU-accelerated workloads.

## E  ABLATION STUDY

In this section, we conduct a series of ablation experiments to analyze the impact of various hyperparameters on the experimental results. Appendix E.1 investigates the effect of $T$. Appendix E.2 examines the impact of $r_s$. Appendix E.3 studies the influence of $K_t$. Appendix E.4 reports the performance under different $K_e$. Appendix E.5 presents the training dynamics of various models. Appendix E.6 evaluates the effect of $\eta$. Appendix E.7 compares vanilla diffusion and MLP. Unless otherwise specified, the default parameters in this section are set as: $N_e = 10000, r_s = 20\%, K_e = 32, K_t = 16, \eta = 0$.

### E.1  DIFFERENT DIFFUSION STEPS

Recent studies (Song et al., 2020a) have pointed out that the number of diffusion steps can significantly affect the quality of the generated solutions. In addition, according to (6), different values of $T$ introduce different levels of noise into the sampling process, which also affects the discretization error in Lemma 1.

To investigate the effect of diffusion steps $T$ on performance, we visualize the training dynamics of DiOpt under different diffusion steps in Figure 5. As shown in the figure, for the four tasks considered, the performances of $T = 10$ and $T = 20$ vary across tasks. Only $T = 100$ consistently achieves the smallest optimality gap across all tasks. Furthermore, for all values of $T$, a significant improvement in feasibility is observed immediately after the bootstrapping phase begins (around 2000 epochs). This observation is consistent with the conclusions drawn in the main text.

### E.2  DIFFERENT SUPERVISED RATIOS

Here $r_s$ critically determines how closely DiOpt approaches the optimal value before bootstrapping initialization. It should be noted that due to the curse of dimensionality (Bellman et al., 1957), the sampling capability of Diffusion-based methods under limited computational resources cannot guarantee sufficient exploration of the solution space. This limitation often leads to suboptimal objective function performance when supervision is inadequate.

As demonstrated in Figure 6, models with $r_s = 0.05, 0.02$, and $0.2$ achieve comparable feasibility performance. However, across both QP and QPSR tasks(with Dimension 100), the $r_s = 0.02$ configuration demonstrates relatively inferior objective function performance compared to $r_s = 0.05, 0.2$, attributable to insufficient supervised learning. In contrast, both $r_s = 0.05, 0.2$ maintain objective function values comparable to Diffusion while achieving satisfactory feasibility.

These results highlight the importance of selecting an appropriate supervised ratio for DiOpt. In this work, we set the default $r_s$ to 0.2 to ensure: (1) model convergence before bootstrapping, and (2) a clear demonstration of effectiveness of bootstrapping through performance improvements.

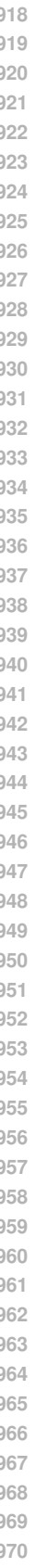

(a) ACOPF57

**ACOPF57**

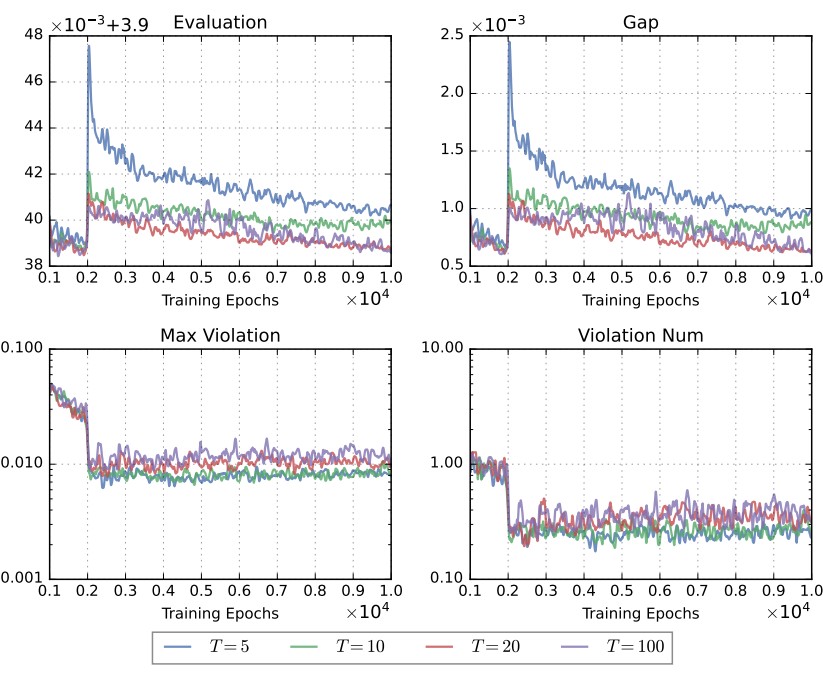

(b) QP

**QP**

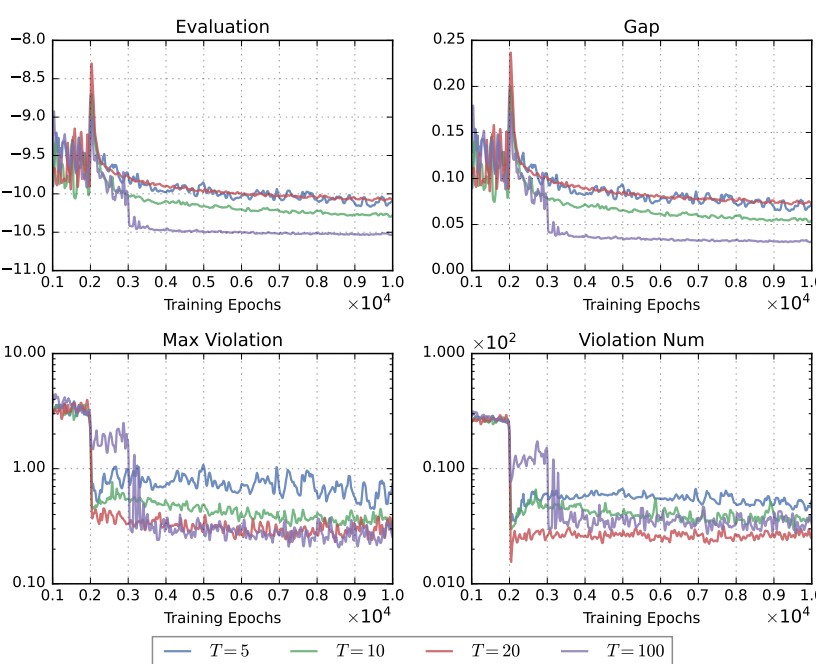

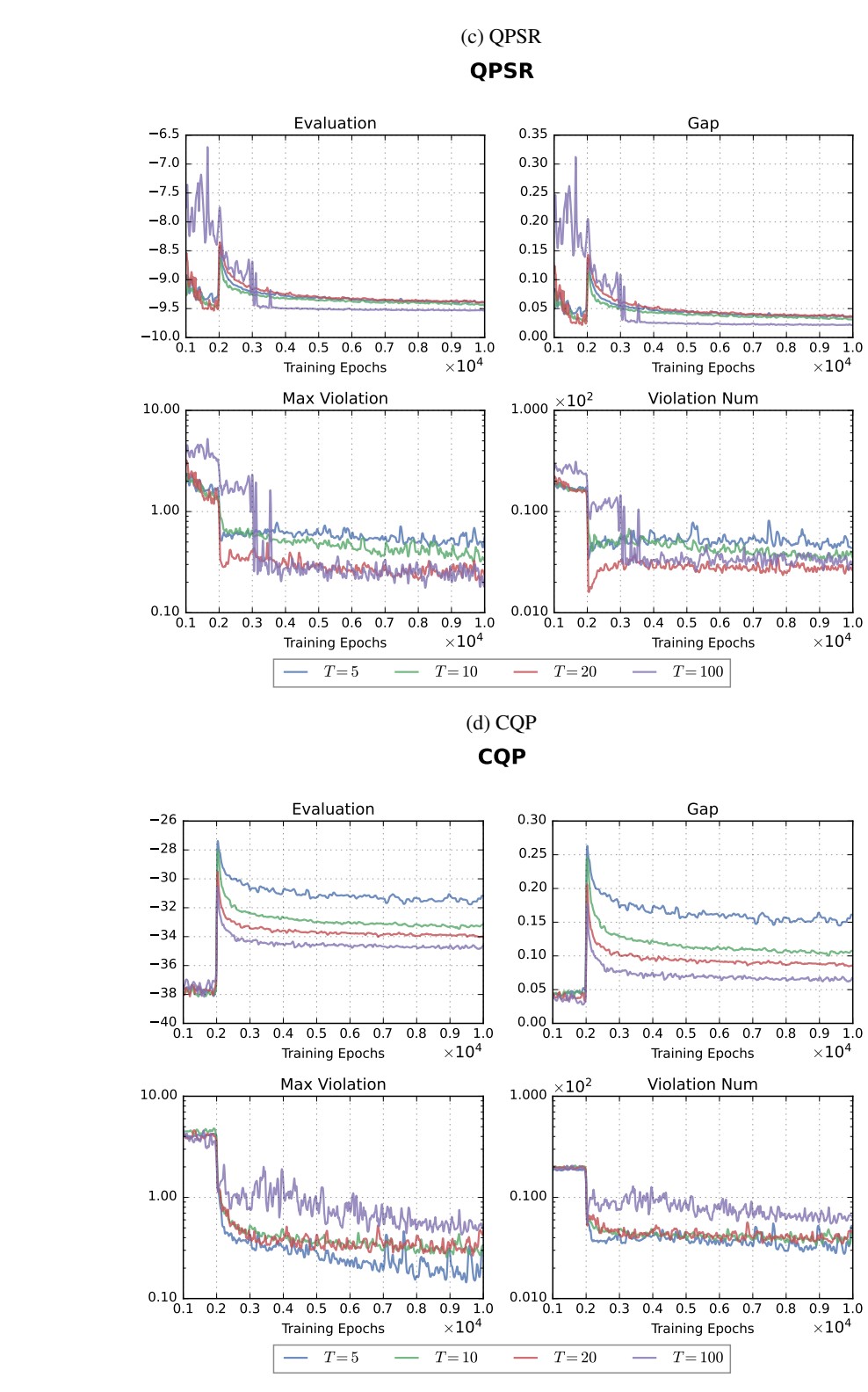

Figure 5: **Effects of Different Diffuion Steps**. In this experiment, we examine how varying the number of diffusion steps $T$ affects performance. All other hyperparameters are fixed as follows: $r_s = 0.2$, $K_t = 16$, and $K_e = 32$ for all values of $T$.

(a) QPSR

**QPSR**

(b) ACOPF57

**ACOPF57**

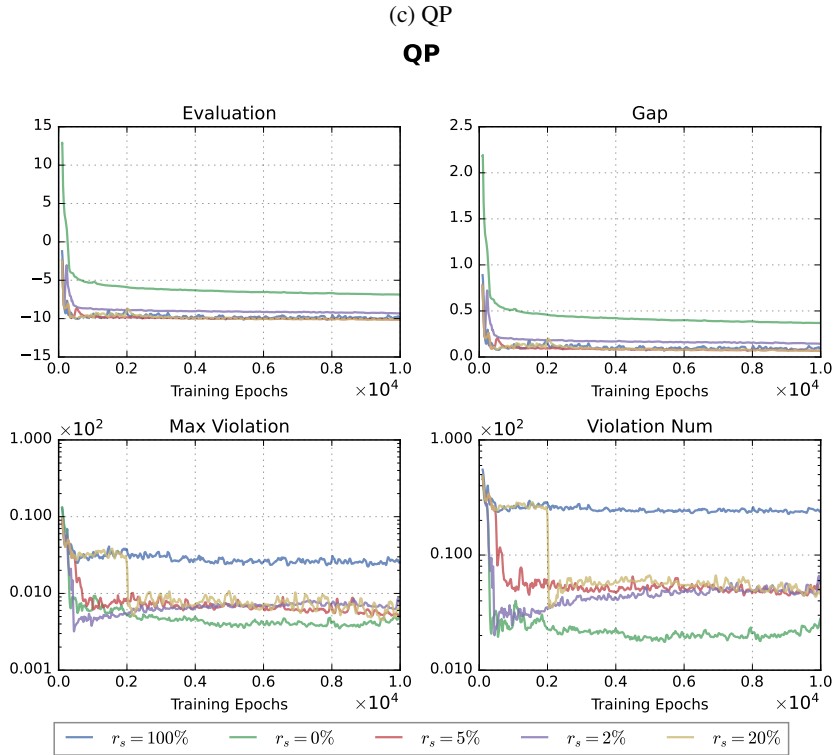

Figure 6: **Effects of Different Supervised Ratio**. In this experiment, we examine how varying the number of supervised ratio $r_s$ affects performance. All other hyperparameters are fixed as follows: $T = 5$, $K_t = 16$, and $K_e = 32$ for all values of $r_s$.

### E.3 DIFFERENT TRAINING SAMPLES

During the training phase of DiOpt, we set $K_t = 1$ for the supervised learning stage. Once bootstrapping begins, $K_t$ is set to 16. Figure 4 demonstrates that DiOpt maintains performance parity with Diffusion prior to bootstrapping, confirming that no additional $K_t$ adjustment is required during supervised learning.

Figure 7 reveals that increased $K_t$ values yield improved objective function performance at the cost of slight feasibility degradation. This trade-off emerges because larger $K_t$ values concentrate the sampling distribution of model nearer to optimal points, consequently narrowing the feasible region. However, given the negligible impact on overall feasibility, we establish $K_t = 16$ as the default configuration of DiOpt. The decision against higher $K_t$ values stems from computational overhead, as they would proportionally increase both training duration and GPU memory requirements.

(a) QPSR

**QPSR**

(b) ACOPF57

**ACOPF57**

(c) QP

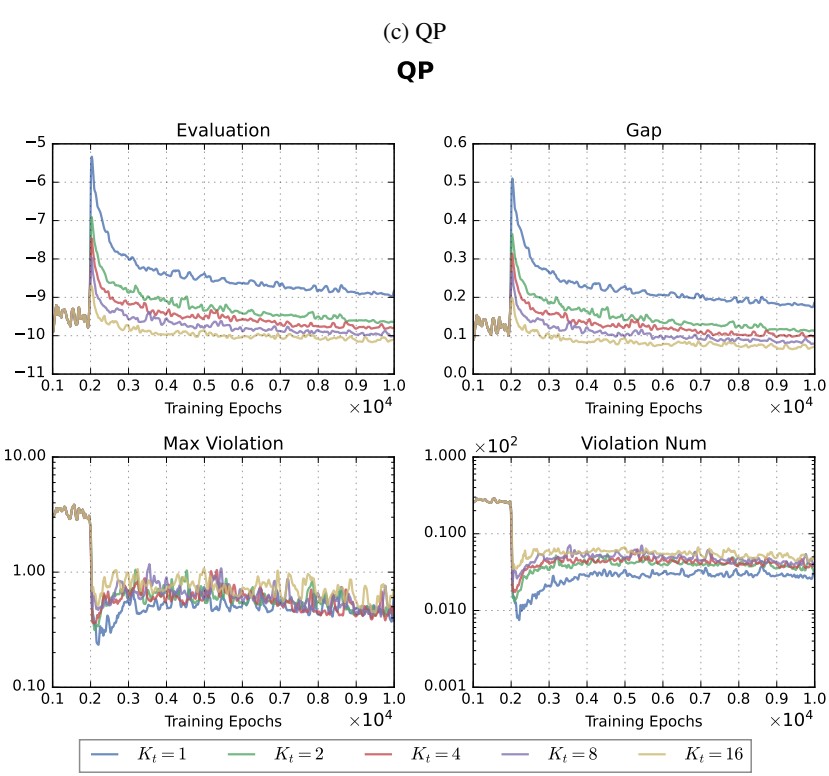

Figure 7: **Effects of different Training Samples**. In this experiment, we examine how varying the number of training samples $K_t$ affects performance. All other hyperparameters are fixed as follows: $r_s = 0.2$, $T = 5$, and $K_e = 32$ for all values of $K_t$.

E.4 DIFFERENT EVALUATION POINTS

Here, we report the performance of Diffusion and DiOpt under different values of $K_e$. It can be observed that, in Figure 8b, Diffusion fails to obtain feasible solutions even when increasing $K_e$. In contrast, as observed in Figure 8a, DiOpt is able to generate feasible solutions while maintaining a small optimality gap in tasks with higher constraint dimensions. This demonstrates that the bootstrapping-based training enables the diffusion model to learn the location of the feasible region.

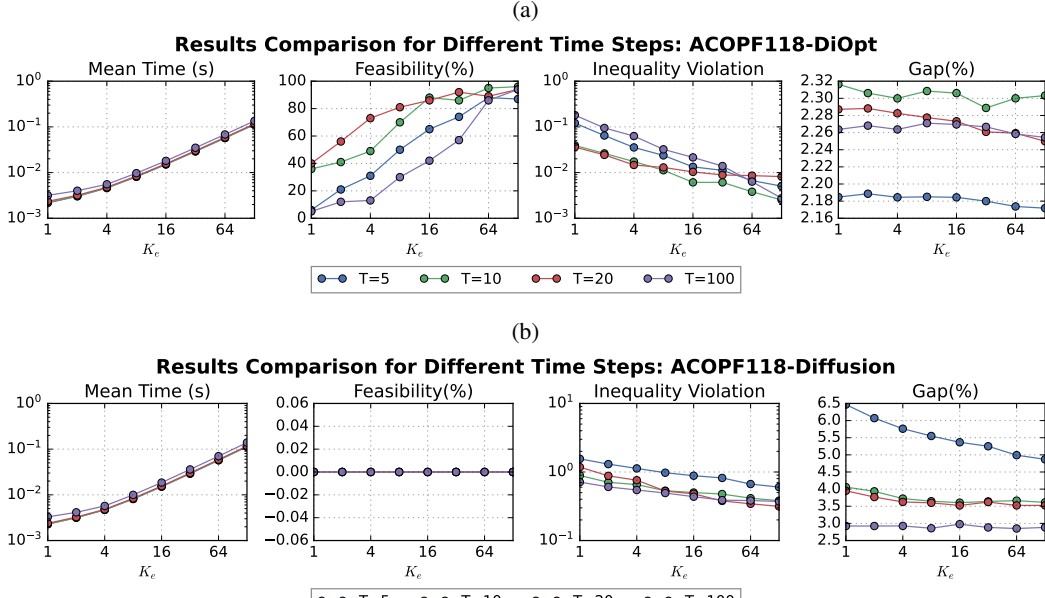

Figure 8: **Effects of number of sampling samples**. In this experiment, we examine how varying the number of sampling points $K_e$ affects performance with $\eta = 1$.

E.5 DIFFERENT TRAINING PROCEDURES

In this subsection, we primarily analyze the training dynamics of DC3, Diffusion, and DiOpt in the main text. It is worth reiterating that the term "Diffusion" here is equivalent to DiOpt with $r_s = 1$. For all experiments in this section, hyperparameters are uniformly set as $K_e = 32$, $K_t = 16$, $r_s = 0.2$, and $T = 5$. Notably, the noise level $\eta$ is configured as 0 in all figures presented, a setting that significantly impacts performance outcomes. This aspect will be further elaborated in Section E.6.

From the Figure 9a and 9b analyses, we observe that both DiOpt and Diffusion exhibit superior performance gaps compared to DC3 throughout the training process, while maintaining comparable constraint violation levels. For the CQP problem (Detailed in Appendix F.5), DC3 exhibits particularly poor performance due to the inherent characteristics of its objective function, where larger $y$ values correspond to steeper gradients. This gradient amplification phenomenon causes DC3 to deviate progressively from feasible regions during iterative updates, ultimately resulting in severe feasibility violations and substantial optimality gaps. In contrast, both Diffusion and DiOpt demonstrate remarkable resilience to such gradient interference, suggesting that diffusion-based optimization frameworks are inherently more robust against sampling point divergence caused by high-gradient objective functions.

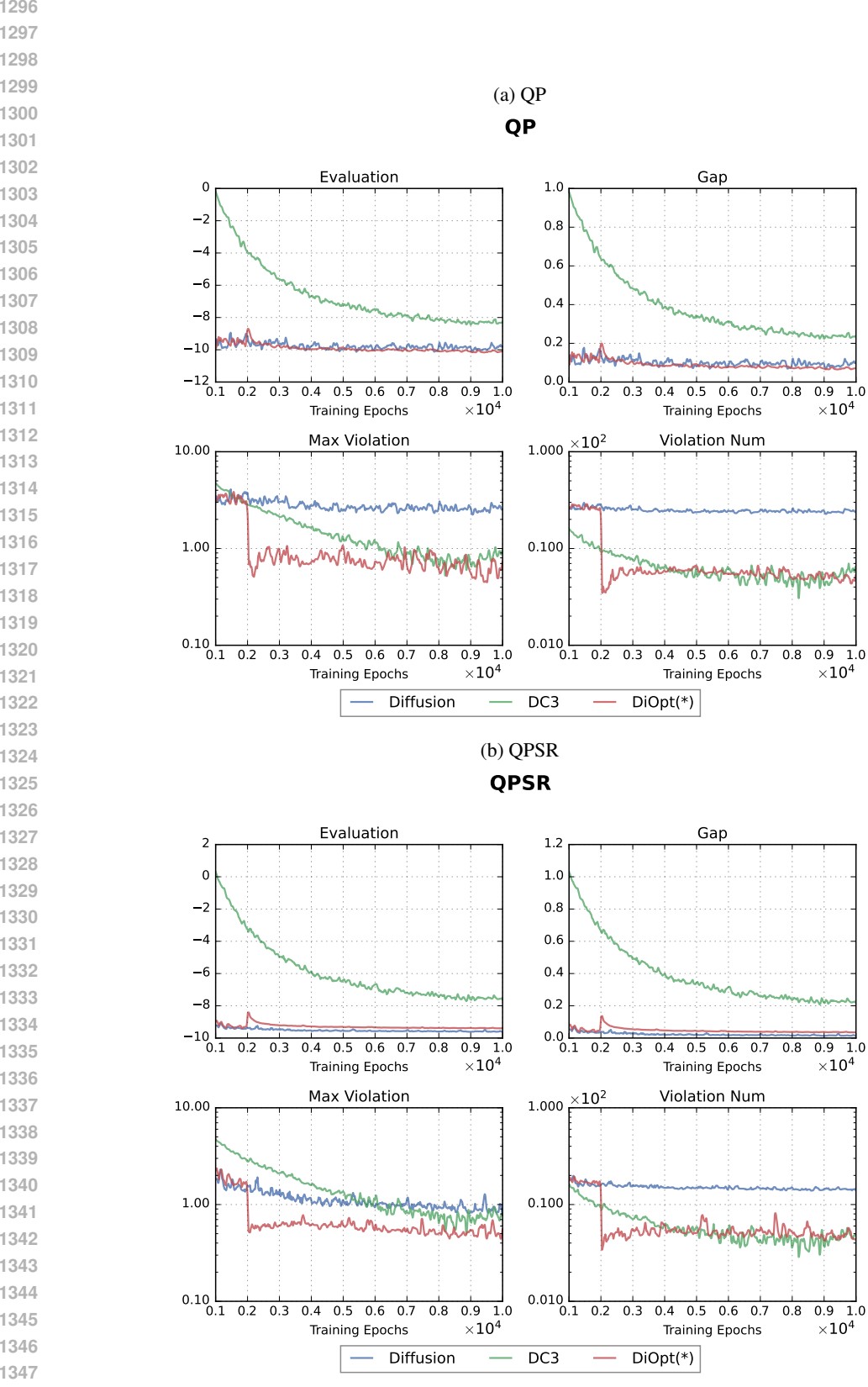

(a) QP

(b) QPSR

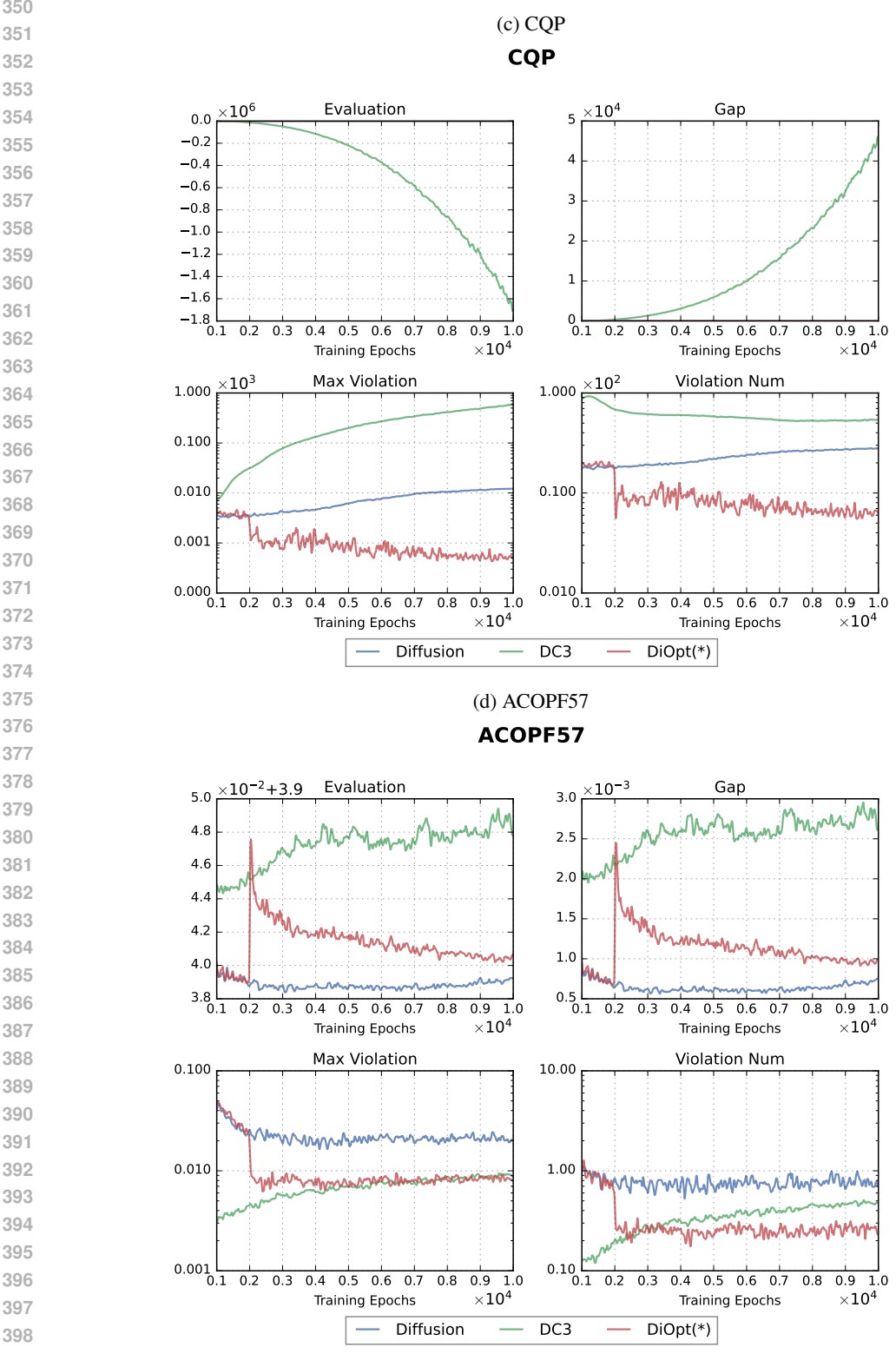

Figure 9: **Training Procedure of DiOpt and Diffusion.** In this experiment, all hyperparameters are fixed as follows: $r_s = 0.2$, $T = 5$, $K_t = 16$, and $K_e = 32$.

### E.6 DIFFERENT NOISE LEVELS

In this section. we investigate the influence of noise level $\eta$ on feasibility rates. As previously discussed, varying $\eta$ determines the "exploration range" of the diffusion model, which significantly impacts its performance on optimization tasks. As shown in Figure 10 and Figure 11, setting $\eta = 0$ implies that even varying $K_e$ does not affect the results. It can be observed that the effect of $\eta$ is not entirely consistent across different values of $K_e$. As shown in the figure, there exists an optimal interval of $\eta$ within which feasibility is maximized. When $\eta$ lies close to this optimal interval, the feasibility performance is best; conversely, when $\eta$ deviates from this interval, the feasibility deteriorates. The location of this optimal interval varies with $K_e$. For instance, when $K_e = 256$, the optimal interval is approximately $1.5 < \eta < 2$. In contrast, for $K_e = 1$, the optimal threshold is attained at $\eta = 0$. Considering resource consumption, setting $K_e = 64$ with $\eta \in [0.9, 1.2]$ is empirically sufficient.

Figure 10: Feasibility of DiOpt under different $\eta$ and $K_e$.

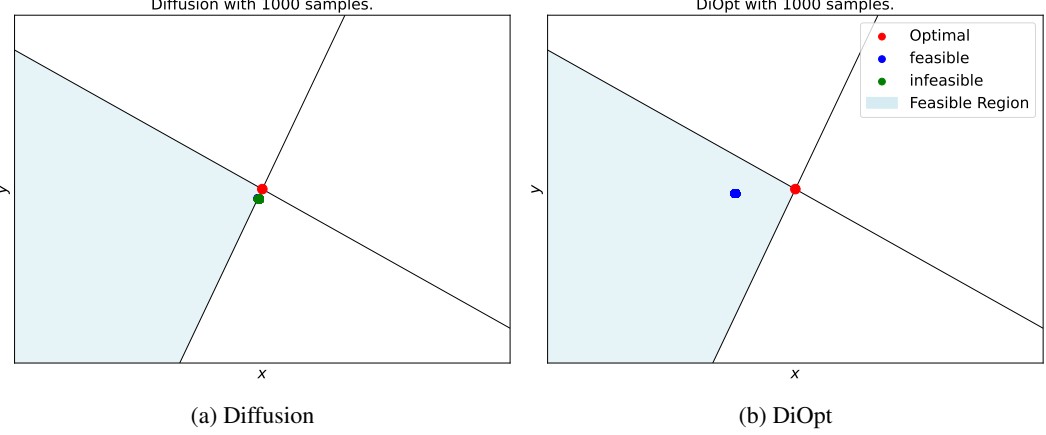

(a) Diffusion        (b) DiOpt

Figure 11: For the $\eta = 0$ configuration in the toy example, observations reveal that despite initial sampling starting from Gaussian-distributed random noise, all trajectories converge to nearly identical positions after denoising when $\eta = 0$. This demonstrates the necessity of noise injection during the sampling process.

### E.7 DIFFUSION V.S. MLP

We conducted a comparison between Vanilla Diffusion (trained by (7)) and a Multi-Layer Perceptron (MLP) to highlight the superiority of diffusion models over simple MLPs in solving optimization problems. The MLP employs the same backbone network as DC3 described in the main text, while the hyperparameters of Vanilla Diffusion are kept consistent with DiOpt in Table 2. For both methods, equation completion was used to handle equality constraints. It can be observed that in most tasks, Diffusion achieves superior performance in terms of both constraint violation and gap. This advantage stems directly from the solution diversity inherent in Diffusion.

Table 6: Comparison between Vanilla Diffusion and MLP. Here, Diffusion refers to the supervised diffusion model

| Problem | Method | Feasibility(%)↑ | Gap(%)↓ | Objective↓ | Time(s)↓ | Ineq Mean↓ | Ineq Max↓ | Ineq Num Viol↓ |
|---|---|---|---|---|---|---|---|---|
| ACOPF57 | Diffusion | $81.33\% \pm 0.00$ | $0.19\% \pm 0.71$ | $3.81 \pm 0.64$ | $0.02 \pm 0.00$ | $0.00 \pm 0.00$ | $0.01 \pm 0.01$ | $0.26 \pm 0.56$ |
| | MLP | $21.00\% \pm 0.00$ | $0.31\% \pm 0.90$ | $3.81 \pm 0.63$ | $0.00 \pm 0.00$ | $0.00 \pm 0.00$ | $0.07 \pm 0.07$ | $1.33 \pm 1.02$ |
| ACOPF118 | Diffusion | $0.00\% \pm 0.00$ | $2.90\% \pm 0.21$ | $13.49 \pm 1.25$ | $0.07 \pm 0.00$ | $0.00 \pm 0.00$ | $0.38 \pm 0.14$ | $9.10 \pm 1.30$ |
| | MLP | $0.00\% \pm 0.00$ | $2.80\% \pm 0.30$ | $13.47 \pm 1.23$ | $0.00 \pm 0.00$ | $0.00 \pm 0.00$ | $0.17 \pm 0.13$ | $7.33 \pm 2.19$ |
| QP | Diffusion | $0.00\% \pm 0.00$ | $0.97\% \pm 3.22$ | $-10.81 \pm 0.55$ | $0.01 \pm 0.00$ | $0.01 \pm 0.03$ | $0.58 \pm 0.65$ | $14.02 \pm 4.20$ |
| | MLP | $0.00\% \pm 0.00$ | $230.15\% \pm 318.77$ | $14.09 \pm 34.49$ | $0.00 \pm 0.00$ | $1.90 \pm 1.86$ | $19.32 \pm 14.21$ | $65.32 \pm 24.62$ |
| QPSR | Diffusion | $0.00\% \pm 0.00$ | $0.43\% \pm 2.87$ | $-9.73 \pm 0.51$ | $0.01 \pm 0.00$ | $0.01 \pm 0.02$ | $0.34 \pm 0.48$ | $11.84 \pm 2.82$ |
| | MLP | $0.00\% \pm 0.00$ | $296.52\% \pm 347.94$ | $19.08 \pm 33.31$ | $0.00 \pm 0.00$ | $1.87 \pm 1.68$ | $19.39 \pm 13.03$ | $64.58 \pm 24.35$ |
| CQP | Diffusion | $0.00\% \pm 0.00$ | $1.47\% \pm 0.85$ | $-36.68 \pm 0.71$ | $0.01 \pm 0.00$ | $0.00 \pm 0.00$ | $0.33 \pm 0.18$ | $11.27 \pm 1.90$ |
| | MLP | $0.00\% \pm 0.00$ | $193.33\% \pm 125.76$ | $-108.99 \pm 46.40$ | $0.00 \pm 0.00$ | $1.12 \pm 0.49$ | $32.82 \pm 13.47$ | $69.88 \pm 9.00$ |
| RETARGETING | Diffusion | $100.00\% \pm 0.00$ | $1.24\% \pm 2.52$ | $1.74 \pm 0.50$ | $0.01 \pm 0.00$ | $0.00 \pm 0.00$ | $0.00 \pm 0.00$ | $0.00 \pm 0.00$ |
| | MLP | $89.66\% \pm 0.00$ | $30.53\% \pm 52.41$ | $2.11 \pm 0.56$ | $0.00 \pm 0.00$ | $0.00 \pm 0.00$ | $0.02 \pm 0.06$ | $0.10 \pm 0.30$ |

## F BENCHMARK CONFIGURATION

In this section, we provide detailed descriptions of some tasks referenced in the main text. These include concave quadratic optimization problems (CQP), relatively complex nonconvex optimization problems with practical significance, and others.

### F.1 BENCHMARK DETAILS

We present a table summarizing the relevant parameters for each benchmark used in Section 6:

Table 7: Benchmark parameters used in our experiments.

| **Problem** | $d_x$ | $d_y$ | $d_z$ | $m$ | $n$ | training set | validation set | test set | active constraints |
|---|---|---|---|---|---|---|---|---|---|
| QP | 50 | 100 | 50 | 250 | 50 | 8334 | 833 | 833 | 30.30(2.03) |
| QPSR | 50 | 100 | 50 | 250 | 50 | 8334 | 833 | 833 | 24.18(2.01) |
| CQP | 50 | 100 | 50 | 250 | 50 | 8334 | 833 | 833 | 51.55(1.18) |
| ACOPF57 | 114 | 128 | 13 | 142 | 114 | 1000 | 100 | 100 | 6.01(1.76) |
| ACOPF118 | 236 | 344 | 107 | 452 | 236 | 1000 | 100 | 100 | 22.70(2.49) |
| Retargeting | 39 | 19 | 19 | 39 | 0 | 1447 | 145 | 145 | 0.97(0.18) |

According to DC3 (Donti et al., 2021), the neural network produces an output vector $\boldsymbol{z} \in \mathbb{R}^{d_z}$. This output is then completed via equality constraints to form the optimization variable

$$y = [\phi_x^\top(z), z^\top]^\top \in \mathbb{R}^{d_y},$$

which satisfies the equality constraints $h(y; x) = 0$. The column 'Active Constraints' reports the mean number of inequality constraints that are active ($\exists i$ s.t. $g_i(y; x) = 0$) in our dataset for each benchmark. These inequality constraints must be directly satisfied by the neural network, with certain constraints enforced by interpolating between their minimum and maximum values (like box constraints in Retargeting).

### F.2 TOY EXAMPLE

The Toy Example shown in Figure 2 is defined as follows:

$$
\begin{aligned}
\min_{y_1, y_2} \quad & f(y_1, y_2) = \left(y_1 - \frac{65}{19}\right)^2 + \left(y_2 - \frac{24}{19}\right)^2 \\
\text{s.t.} \quad & -4y_1 - 3y_2 \leq -12, \\
& -y_2 \leq 0, \\
& 4y_1 + 5y \leq 20, \\
& 0 \leq y_1, y_2 \leq 5.
\end{aligned}
\tag{19}
$$

The Optimal Point (point marked as red) in Figure 2 is $\left(\frac{65}{19}, \frac{24}{19}\right)$.

### F.3 QUADRATIC PROGRAMMING (QP)

The Simple Problem discussed in the text is defined as follows:

$$
\begin{aligned}
\min_{y \in \mathbb{R}^n} \quad & \frac{1}{2}y^T Q y + p^T y, \\
\text{s.t.} \quad & Ay = x, \\
& Gy \le h.
\end{aligned}
\tag{20}
$$

Here, $x$ is treated as a conditional parameter of the optimization problem, sampled uniformly from $[-1, 1]$ across all instances. $Q$, $p$, $A$, and $h$ remain fixed. $Q$ is a diagonal matrix whose diagonal elements are independently and identically sampled from $[0, 1]$. The vector $p$ is generated using the same method as $Q$. Elements of matrices $A$ and $G$ are sampled from a standard normal distribution. To ensure the feasibility of $Gy \le h$, $h$ is constructed as:

$$
h_i = \sum_j |(GA^+)_{ij}|,
\tag{21}
$$

where $A^+$ denotes the pseudoinverse of $A$. The optimal solutions are generated through IPOPT (Wächter & Biegler, 2006). **10000 examples have been generated for this task.**

### F.4 QUADRATIC PROGRAMMING WITH SINE REGULARIZATION (QPSR)

The Nonconvex Problem in the text is formulated as:

$$
\begin{aligned}
\min_{y \in \mathbb{R}^n} \quad & \frac{1}{2}y^T Q y + \alpha \cdot p^T \sin(y), \\
\text{s.t.} \quad & Ay = x, \\
& Gy \le h.
\end{aligned}
\tag{22}
$$

Here, $x$ similarly serves as a conditional parameter, and the generation methods for $x$, $Q$, $p$, $A$, $G$, and $h$ align with those in the Simple Problem. In our experiment, $\alpha$ was setted as 1. The optimal solutions are generated through IPOPT. **10000 examples have been generated for this task.**

### F.5 CONCAVE QUADRATIC PROGRAMMING (CQP)

The CQP discussed in the text is defined as follows:

$$
\begin{aligned}
\min_{y \in \mathbb{R}^n} \quad & \frac{1}{2}y^T Q y + p^T y, \\
\text{s.t.} \quad & Ay = x, \\
& Gy \le h.
\end{aligned}
\tag{23}
$$

Here, $x$ is treated as a conditional parameter of the optimization problem, sampled uniformly from $[-1, 1]$ across all instances. $Q$, $p$, $A$, and $h$ remain fixed. $Q$ is a diagonal matrix whose diagonal elements are independently and identically sampled from $[-1, 0]$. The vector $p$ is generated using the same method as $Q$. Elements of matrices $A$ and $G$ are sampled from a standard normal distribution. To ensure the feasibility of $Gy \le h$, $h$ is constructed as:

$$
h_i = \sum_j |(GA^+)_{ij}|,
\tag{24}
$$

where $A^+$ denotes the pseudoinverse of $A$. The optimal solutions are generated through IPOPT. **10000 examples have been generated for this task.** It is important to note that this task is deliberately designed to challenge gradient-based methods. Specifically, it exhibits the property

$$\lim_{\|y\|\to\infty} \|\nabla f(y;x)\| = \infty,$$

which implies that the gradient norm diverges as $\|y\|$ grows. Consequently, methods such as DC3, which rely on $\nabla f(y;x)$ to train the neural network, fail on this problem.

### F.6 ACOPF

The AC Optimal Power Flow (ACOPF) (Cain et al., 2012; Shi et al., 2017) is a core problem in power systems, aiming to minimize generation costs by adjusting active/reactive power outputs of generators, voltage magnitudes, and phase angles while satisfying constraints such as power balance, line flow limits, and voltage limits. Although the generation costs are merely simple quadratic functions, the intricate constraints render the ACOPF a highly non-convex problem. This results in traditional solution algorithms for ACOPF encountering issues such as global optimality and excessive computation times etc. Recent studies have proposed relaxation approaches (Bingane et al., 2018) and machine learning-based approaches (Zamzam & Baker, 2020; Zhang & Zhang, 2022; Jiang et al., 2024; Zhao & Barati, 2024) to address these issues.

More specifically, an ACOPF problem involves $N$ nodes, including load buses $\mathcal{L}$, a reference bus $\mathcal{R}$, and generator buses $\mathcal{G}$. Variables include active power $p_g$, reactive power $q_g$, active demand $p_d$, reactive demand $q_d$, voltage magnitude $|v|$, and voltage phase angle $\theta$. Load buses (representing non-generating nodes) satisfy $(p_g)_{\mathcal{L}} = (q_g)_{\mathcal{L}} = 0$. The reference bus provides a phase angle reference, with $\theta_{\mathcal{R}} = \theta_{\text{ref}}$. Network parameters are described by the admittance matrix $Y$. The ACOPF is formalized as follows, where $v = |v|e^{i\theta}$, and $A, b$ are fixed parameters related to generation costs:

$$
\begin{aligned}
\min_{p_g, q_g, v, \theta} \quad & p_g^T A p_g + b^T p_g, \\
\text{s.t.} \quad & \underline{p}_g \le p_g \le \overline{p}_g, \\
& \underline{q}_g \le q_g \le \overline{q}_g, \\
& \underline{|v|} \le |v| \le \overline{|v|}, \\
& \theta_{\mathcal{R}} = \theta_{\text{ref}}, \\
& (p_g)_{\mathcal{L}} = (q_g)_{\mathcal{L}} = 0, \\
& (p_g - p_d) + i(q_g - q_d) = \text{diag}(v)Yv^*.
\end{aligned}
\tag{25}
$$

where $A, b$ represent as the cost coefficient, the underline represent the lower bound, and overline represent the upper bound. In this formulation, nodal demands $p_d$ and $q_d$ act as conditional parameters. Our experiments test the 57-bus system (ACOPF57) and 118-bus system (ACOPF118), with optimal solutions obtained via IPOPT (Wächter & Biegler, 2006). **1200 problems are generated for both ACOPF57 and ACOPF118**.

### F.7 RETARGETING PROBLEM

The motion retargeting task can be formulated as an optimization problem, where the objective is to minimize the discrepancy between the motion of the SMPL human model and the H1 robot model. This task involves not only the alignment of joint positions but also the consideration of differences in kinematic structure, body proportions, joint alignment, and end-effector positioning. Due to the significant differences between the kinematic structure of the SMPL model and the kinematic tree of the H1 humanoid robot, He et al. (2024) proposed a two-step method for preliminary motion retargeting. In the first step, given that the body shape parameters $\beta$ of the SMPL human model can represent a variety of body proportions, we optimize to find a body shape $\beta'$ that best matches the humanoid robot's structure, thereby minimizing the joint position discrepancies between the models. This ensures that the joint positions of the SMPL model and H1 robot align as closely as possible, laying the foundation for subsequent retargeting.

Once the optimal $\beta'$ is determined, the second step involves mapping the joint positions and postures of the H1 robot to their corresponding positions in the SMPL model using forward kinematics. This process takes into account the kinematic constraints of the robot, ensuring the validity of joint positions. Finally, to further refine the joint alignment, we minimize the differences in the positions of 11 key joints, adjusting the joint configuration between the SMPL model and the H1 robot. It is important to note that the retargeting process goes beyond adjusting joint positions—it also involves the alignment of end-effectors (such as ankles, elbows, and wrists). Special attention is given to the precise alignment of these key points to ensure that the human motion is smoothly transferred to the humanoid robot. Given a sequence of motions expressed in SMPL parameters, which takes as input the joint positions $\boldsymbol{P}_{SMPL}$, root rotation $\boldsymbol{R}_{root}$, and transform offset $\boldsymbol{O}_{offset}$ from the SMPL model, and computes the global joint positions $\boldsymbol{P}_{H1}$ of the H1 robot model using forward kinematics. The loss function is defined as the difference between the computed H1 joint positions and the corresponding SMPL joint positions. The optimization problem is defined as follows:

$$
\begin{aligned}
\min_{\boldsymbol{P}_{H1}} \quad & \|\text{FK}(\boldsymbol{P}_{H1}, \boldsymbol{R}_{root}, \boldsymbol{O}_{offset}) - \boldsymbol{P}_{SMPL}\|_2^2 + \lambda\|\boldsymbol{P}_{H1}\|_2^2, \\
\text{s.t.} \quad & \boldsymbol{P}_{lower} \leq \boldsymbol{P}_{H1} \leq \boldsymbol{P}_{upper}, \\
& \|\boldsymbol{P}_{H1}\|_2^2 \leq 4.
\end{aligned}
\tag{26}
$$

The $\ell_2$ norm penalty ensures smoother values and prevents $\boldsymbol{P}_{H1}$ from becoming excessively large during optimization. Large control inputs could be impractical and could even damage the robot hardware. **1737 examples have been generated for this task via IPOPT**.

# G    BASELINE SETTINGS

## G.1    DC3

DC3 (Deep Constraint Completion and Correction) is a neural network-based constrained optimization solver. Unlike direct prediction of solutions via neural networks, DC3 incorporates two key components: an *equality completion* operator $\varphi_{\boldsymbol{x}}$ and an *inequality correction* operator $\rho_{\boldsymbol{x}}$. These mechanisms significantly improve the feasibility of the obtained solutions.

Moreover, DC3 adopts a self-supervised learning paradigm. Instead of requiring optimal solutions as supervision, it constructs its loss function as follows:

$$
\ell_{\text{soft}}(\hat{\boldsymbol{y}}) = f(\hat{\boldsymbol{y}}; \boldsymbol{x}) + \lambda_g \cdot \|\text{ReLU}(g(\hat{\boldsymbol{y}}; \boldsymbol{x}))\|_2^2 + \lambda_h \cdot \|h(\hat{\boldsymbol{y}}; \boldsymbol{x})\|_2^2,
$$

where $f(\cdot; \boldsymbol{x})$ denotes the objective function, while $g(\cdot; \boldsymbol{x})$ and $h(\cdot; \boldsymbol{x})$ represent inequality and equality constraints, respectively. The ReLU-based term penalizes constraint violations.

The training and sampling procedures of DC3 are outlined in Algorithm 2, adapted from (Donti et al., 2021). In our experiments, we fix $\lambda_g = \lambda_h = 5$. In this work, we define $N_\theta$ as a three-layer neural network comprising two hidden layers (each with 512 neurons, ReLU activation, batch normalization, and dropout with $p = 0.2$).

## G.2    MLP

In our experiments, we include an MLP baseline that shares the **identical network architecture** with DC3. However, the MLP differs in two key aspects: 1.) It only employs equality completion $\phi_{\boldsymbol{x}}$ and omits inequality correction $\rho_{\boldsymbol{x}}$. 2.) During training, it directly minimizes the MSE loss between predictions and ground-truth solutions:

$$
\ell_{\text{mse}} = \|\widetilde{\boldsymbol{y}} - \boldsymbol{y}^\star\|_2^2.
$$

## G.3    MODEL-BASED DIFFUSION

We attempt to adopt the model-based diffusion method proposed in (Pan et al., 2024) as a baseline for this work. Since practical application scenarios may not strictly satisfy the conditions specified in the original paper, we adapt the method with specific modifications. Concretely, when calculating the probability score for each sample, we compute it as follows:

$$
p_i = \text{P}(\boldsymbol{y}_i | \boldsymbol{x}) := f(\boldsymbol{y}_i; \boldsymbol{x}) + \lambda_h \|h(\boldsymbol{y}_i; \boldsymbol{x})\|_2 + \lambda_g \|\text{ReLU}(g(\boldsymbol{y}_i; \boldsymbol{x}))\|_2.
\tag{27}
$$

---

**Algorithm 2** Deep Constraint Completion and Correction (DC3)

---

1: **Assume:** Equality completion procedure $\varphi_{\boldsymbol{x}} : \mathbb{R}^{d_{\boldsymbol{y}} - d_{\text{neq}}} \to \mathbb{R}^{d_{\text{neq}}}$
2: **procedure** TRAIN $(X)$
3: Initialize neural network $N_\theta : \mathbb{R}^{d_{\boldsymbol{x}}} \to \mathbb{R}^{d_{\boldsymbol{y}} - d_{\text{neq}}}$
4: **while** not converged **do**
5:   **for** $\boldsymbol{x} \in X$ **do**
6:     Compute partial variables: $\boldsymbol{z} = N_\theta(\boldsymbol{x})$
7:     Complete to full variables: $\tilde{\boldsymbol{y}} = \begin{bmatrix} \boldsymbol{z}^T & \varphi_{\boldsymbol{x}}(\boldsymbol{z})^T \end{bmatrix}^T \in \mathbb{R}^{d_{\boldsymbol{x}}}$
8:     Correct to feasible (or approx. feasible) solution: $\hat{\boldsymbol{y}} = \rho_{\boldsymbol{x}}^{(train)}(\tilde{\boldsymbol{y}})$
9:     Compute constraint-regularized loss: $\ell_{\text{soft}}(\hat{\boldsymbol{y}})$
10:     Update $\theta$ using $\nabla_\theta \ell_{\text{soft}}(\hat{\boldsymbol{y}})$
11:   **end for**
12: **end while**
13: **end procedure**
14: **procedure** TEST $(\boldsymbol{x}, N_\theta)$
15: Compute partial variables: $\boldsymbol{z} = N_\theta(\boldsymbol{x})$
16: Complete to full variables: $\tilde{\boldsymbol{y}} = \begin{bmatrix} \boldsymbol{z}^T & \varphi_{\boldsymbol{x}}(\boldsymbol{z})^T \end{bmatrix}^T$
17: Correct to feasible solution: $\hat{\boldsymbol{y}} = \rho_{\boldsymbol{x}}^{(test)}(\tilde{\boldsymbol{y}})$
18: **Return** $\hat{y}$
19: **end procedure**

---

Here, $\boldsymbol{y}_i$ represents the $i$-th sample within the complete collection of samples in one diffuse step. The subsequent algorithmic steps remain consistent with Algorithm 1 in (Pan et al., 2024). For all experiments, the number of samples is set to 256, the number of diffusion steps is set to 100, and $\lambda_h = \lambda_g = 10$. The specific process of the model-based diffusion with completion is outlined in Algorithm 3, where "Completion" denotes the task-specific completion procedure.

---

**Algorithm 3** Model-based Diffusion with completion

---

**Input:** $\boldsymbol{z}^{(N)} \sim \mathcal{N}(0, I)$, Condition Parameter $\boldsymbol{x}$, Number of Diffusion Steps $N$, Number of Samples $d$.
**for** $i = N$ to $1$ **do**
  Sample $d$ samples $\mathcal{Z}^{(i)} = [\boldsymbol{z}_1^i, ..., \boldsymbol{z}_d^i] \overset{i.i.d}{\sim} \mathcal{N}\left(\frac{\boldsymbol{z}^{(i)}}{\sqrt{\alpha_{i-1}}}, \left(\frac{1}{\bar{\alpha}_{i-1}} - 1\right) I\right)$
  Get completion: $\mathcal{Y}^{(i)} = [\boldsymbol{y}_1^i, ..., \boldsymbol{y}_d^i] = \text{Completion}(\mathcal{Z}^{(i)}; \boldsymbol{x})$
  Calculate probability score: $p_j = P(\boldsymbol{y}_j^i | \boldsymbol{x})$
  Estimate New Center: $\boldsymbol{z}^{(i-1)} = \frac{\sum_{j=1}^{d} p_j \boldsymbol{z}_j^i}{\sum_{j=1}^{d} p_j}$
**end for**
Complete partial solution: $\boldsymbol{y}^{(0)} = \text{Completion}(\boldsymbol{z}^{(0)}; \boldsymbol{x})$
**Return** Optimized solution $\boldsymbol{y}^{(0)}$

---

