# OpenReview forum: "Exploring the Boundary of Diffusion-based Methods for Solving Constrained Optimization"
_ICLR.cc/2026/Conference — Submitted to ICLR 2026_

### Official Review · Reviewer_iA95 · 2025-10-16

**Soundness:** 2
**Presentation:** 1
**Contribution:** 1
**Rating:** 2
**Confidence:** 3

**Summary:**

This paper explores constrained optimization in generative diffusion models. The main problem that the paper tackles is the fact that a diffusion model may end up generating samples outside of the feasible region due to the inherent stochasticity of the generative process. This is indeed an important problem that warrants a rigorous scientific investigation and the paper is timely. To address the problem, the paper proposes DiOpt, which introduces a weighted bootstrapping method to self-supervise the model to generate samples within the feasible region. The idea, as I understand, is to generate samples ("candidate points") during training and weigh them according to constraint violation (feasible points get positive weights and infeasible points negative weights); The diffusion model loss term (squared error of predicted noise) is then weighed accordingly and used for training the diffusion model.

**Strengths:**

The paper addresses an important problem for the community and has the potential to help us overcome some of the current limitations we are facing with diffusion models. The proposed weighting scheme is somewhat new, and the method is validated against other methods on several benchmarks.

**Weaknesses:**

I guess my biggest problem is with the scientific contribution. First and foremost, even though the paper claims to address the hard-constrained generation problem in various places, such as Table 1 and the intro section, in my opinion, this is still a soft-constrained generation problem. If I understood correctly, the model is simply "discouraged" from generating infeasible samples via the bootstrapped weight values during the training stage. I don't find anything that actually hard-constrains the generated samples to be within the feasible region, and their experiment data also shows that (e.g., the first column of Table 2).

However, even aside from the discussion of whether it is a hard-constrained method or not, I don't know if the paper contains a sufficient amount of new scientific knowledge or methodological breakthroughs. The main idea of the DiOpt method is to weigh the loss function via bootstrapped samples so that the generation of feasible samples is promoted, whereas infeasible samples are suppressed. However, considering the high scientific standard of ICLR, I don't know if this is something I would call an "ICLR-worthy" idea. Also, the mathematical analysis in Section 4 on "why ... diffusion models encounter infeasibility issues in constrained optimization," which is argued as a contribution of this paper, is rather underwhelming: It defines feasibility in **linear** programming mostly, and there are a couple of sentences showing an asymptotic bound on the probability of generated sample existing in the feasible region unfortunately without any proofs or rigorous dicussions. So again, I don't know if this is an ICLR-worthy contribution to the scientific community.

Furthermore, overall, the presentation can be improved significantly. First and foremost, the scientific contribution of this work is unclear. Both the intro and the abstract could benefit from jumping straight to the exact problem that the authors want to address, how they propose to solve it, and what contribution to the scientific community they are presenting. Also, the bullets in the intro, which are supposed to summarize their contributions, don't really do justice. That's because these bullets mostly talk about "what" they do, instead of "why" it is important. In its current form, the contribution seems to be very narrowly positioned to the development of this very specific recipe that employs bootstrapping to discourage the model from creating infeasible samples.

The paper could also benefit from more high-level, intuitive explanations of the core scientific idea. Although my research isn't characterized as optimization research, I teach a course on that topic and would claim that I am fairly familiar with the technical content. Despite that, it required me several readings to grasp what the core scientific idea was.

**Questions:**

- Is DiOpt limited only to linear, convex optimization problems, or can it generalize to more complex non-convex optimization problems as well?

---

> ### Author Response · Authors · 2025-11-23
>
> Thanks for your valuable comments and suggestions. Here we address your detailed comments as below:
>
> > **Q1**: Is DiOpt limited only to linear, convex optimization problems, or can it generalize to more complex non-convex optimization problems as well?
>
> **A1**: No, DiOpt is developed to solve **general (include convex and nonconvex) constrained optimization**. The evaluation tasks, including QPSR, CQP, ACOPF57/118, and retargeting are all **nonconvex**. Compared to previous methods, DiOpt significantly improves feasibility and optimality. Especially, ACOPF57/118 are an considerably challenging problem in smart grid [R1, R2, R3]. In that case, there is no doubt that DiOpt can generalize to complex non-convex optimization problem.
>
> [R1] Shi, Y., Tuan, H. D., Tuy, H., & Su, S. (2017). Global optimization for optimal power flow over transmission networks. Journal of Global Optimization, 69(3), 745-760.
>
> [R2] Cain, M. B., O’neill, R. P., & Castillo, A. (2012). History of optimal power flow and formulations. Federal Energy Regulatory Commission, 1, 1-36.
>
> [R3] Gopinath, S., & Hijazi, H. L. (2022, July). Benchmarking large-scale ACOPF solutions and optimality bounds. In 2022 IEEE Power & Energy Society General Meeting (PESGM) (pp. 1-5).
>
>
>
> > **Q2**: I guess my biggest problem is with the scientific contribution. First and foremost, even though the paper claims to address the hard-constrained generation problem in various places, such as Table 1 and the intro section, in my opinion, this is still a soft-constrained generation problem. If I understood correctly, the model is simply "discouraged" from generating infeasible samples via the bootstrapped weight values during the training stage. I don't find anything that actually hard-constrains the generated samples to be within the feasible region, and their experiment data also shows that (e.g., the first column of Table 2).
>
> **A2**: We appreciate the reviewer’s thoughtful comment. In fact, Diopt adopts different strategies to handle hard equality constraints and inequality constraints. For **hard equality constraints**, we follow a standard technique in learn-to-optimize literature to handle them as mentioned in line 177-178 of Section 3. Concretely, we divide variables into basic variables $y_b$ and nonbasic variables $y_n$. The diffusion solver only output the basic variables with the left solved (whith the equation $h(y_b, y_n;x)=0$) by an equation solver. In that case, **equality constraints can be exactly satisfied**. For more details, we have hightlighted them in line 377-381 of the revised version.
>
> As to inequality constraints, we agree that DiOpt cannot guarantee 100\% feasibility in any case. In fact, achieving 100% feasibility has long been a major challenge in the learning-to-optimize field. Thus, our goal is to **improve feasible high-quality solutions under a proper budget** rather than ensure 100\% feasibility.
>
> Thus, what DiOpt contributes to hard inequality constraints is not a guarantee, but a **remedy for the feasibility collapse in all existing diffsuion-based solver**, a failure mode we identify and analyze in Section 4 and Appendix A, B. Empirically, DiOpt improves feasibility dramatically across all tasks (Table 2: $84.33\\%$ on ACOPF118) *without* increased inference cost. This is a meaningful and practically relevant contribution even without 100% feasibility.
>
> In summary, with two different strategies for hard equality and inequality constraints, Diopt can reliably satisfy the constraints in the vast majority of cases and outperforms all existing learning-based optimization methods. Besides, for the few failure cases, we can utilize slow solver to solve them to ensure the feasibility in practice. However, with more feasible solution from DiOpt, we can save much more time in real-world applications to meet real-time requirements.

---

> ### Author Response · Authors · 2025-11-23
>
> > **Q3**: The main idea of the DiOpt method is to weigh the loss function via bootstrapped samples so that the generation of feasible samples is promoted, whereas infeasible samples are suppressed. However, considering the high scientific standard of ICLR, I don't know if this is something I would call an "ICLR-worthy" idea.
>
> **A3**: We believe DiOpt is an "ICLR-worthy" idea since it addresses an essential problem in exsiting diffusion-based optimization methods **that have been published in top conferences including ICLR**. **All existing diffusion-based learn-to-optimize** works such as RectFlow [R4] (accepted at ICLR 2024) and DiffuSolve [R5] (published in L4DC 2025) **follow a supervised-learning paradigm**, which trains the diffusion solver to fit optimal solutions. As we analysed in Section 4, *the final distribution of supervised diffusion is approximately a Gaussian distribution around the optimum, which leads to a large mismatch with the feasible region, especially in high-dimensional tasks*. This analysis explains why prior supervised diffusion solvers tend to generate infeasible samples even if trained with optimal solutions after enough iterations.
>
> In contrast, DiOpt addresses this inherent limitation with the weighted bootstrapping training mechanism, which can enforces diffusion model to generate feasible points with a high probability. Specifically, the self-supervised bootstrapped weighting mechanism can first reshape the diffusion distribution towards the **feasible region** to ensure the feasibility, and then optimize to shift the diffusion distribution close the optimum. This shift enables DiOpt to outperform DC3, MBD, and supervised diffusion on all complex benchmarks. Thus, we believe DiOpt is an "ICLR-worthy" idea.
>
>
> [R4] Liang, E., & Chen, M. (2024). Generative learning for solving non-convex problem with multi-valued input-solution mapping. In The Twelfth International Conference on Learning Representations.
> [R5] Li, A., Ding, Z., Dieng, A. B., & Beeson, R. (2025). DiffuSolve: Diffusion-based Solver for Non-convex Trajectory Optimization. Proceedings of Machine Learning Research, 283, 45-58.
>
> > **Q4**: Also, the mathematical analysis in Section 4 on "why ... diffusion models encounter infeasibility issues in constrained optimization," which is argued as a contribution of this paper, is rather underwhelming: It defines feasibility in linear programming mostly, and there are a couple of sentences showing an asymptotic bound on the probability of generated sample existing in the feasible region unfortunately without any proofs or rigorous dicussions. So again, I don't know if this is an ICLR-worthy contribution to the scientific community.
>
> **A4**: Thanks for raising the concern. First of all, we have provided full **theoretical proofs** in **Appendix A** and **Appendix B** and have illustrated that in line 253-254. Due to the page limitation, we have no choice but to put the full proofs in the appendix. Besides, it is very difficult to directly analyse a general optimization problem since the objective and constraints can be of any form. Therefore, we resort to linear programming to provide a rigorous proof. Moreover, as stated in line 250, the conclusion can be extended to **nonlinear constraints** via local linearization. This extension has been also **empirically validated** in our nonlinear benchmarks (Table 5 in Appendix E.7 which is Table 6 in the revised version).

---

> ### Author Response · Authors · 2025-11-23
>
> > **Q5**: First and foremost, the scientific contribution of this work is unclear;  That's because these bullets mostly talk about "what" they do, instead of "why" it is important.
>
> **A5**: Thanks for your suggestion. We have revised the Introduction and the new contributions are summerized as follows:
>
> - We reveal an inherent limitation of existing diffusion-based methods for optimization: they often produce infeasible solutions. This limitation arises from the fact that existing diffusion-based methods all follow a supervised paradigm, which leads to the mismatch between the target distribution of diffusion solver and the feasible region of the problem to be solved, especially in high-dimensional spaces.
>
> - To handle this inherent limitation, we propose DiOpt as a remedy for all existing diffusion-based methods. By introducing an extra weighted bootstrapping mechanism after supervised learning, the feasibility of the diffusion solver is enhanced while maintaining decent optimality. Notably, DiOpt is the first method that resolves the mismatch limitation and introduces the bootstrapping training mechanism into learning-based optimization methods.
>
> - We evaluate the proposed DiOpt method in a diverse range of nonconvex optimization problems, including synthetic concave QP problems, power grid control, motion retargeting, and wireless power allocation. This comprehensive evaluation encompasses various optimization scenarios with convex and nonconvex objectives and constraints, demonstrating the method’s generalizability across different cases.

---

### Official Review · Reviewer_tjYr · 2025-10-29

**Soundness:** 3
**Presentation:** 3
**Contribution:** 3
**Rating:** 6
**Confidence:** 4

**Summary:**

This paper apply diffusion model on learning to optimize problem to solve constrained optimization problem. Compared to standard supervised diffusion model, which produces infeasible in high dimensional problems, their proposed DiOPt method can hold the constrains better thanks to the combination of supervised and self supervised learning. In self supervised learning stage, a weighted loss is applied to bootstrap feasible samples. The design yields better feasibility and optimality trade-off compared to baselines.

**Strengths:**

- the bootstrapping self-supervised training can effectively bias the sample towards the feasible region and is compatible with different diffusion formulations

- the proposed method is empirically validated on diverse constrained optimization problems with detailed ablation

**Weaknesses:**

- the method introduces extra hyperparameters including weight, reset frequency and lookup updates. It would be beneficial if author and provide heuristics on how to choose those hyperparameters and how sensitive the algorithm are to those parameters.

- the author only report evaluation time in the main text. It would be helpful to also report training time for different method to see the overhead introduced by two-stage training.

**Questions:**

- can DiOPT handle equality constraints?

- how sensitive is DiOPT to parameters like rs and Kt in the paper?

- in standard constrained optimization, Lagrangian multiplier is employed to control the penalty level based on violation. is it possible to use the same practice to make the weight update for constrains less rely on heuristics?

---

> ### Author Response · Authors · 2025-11-23
>
> We would like to thank the Reviewer tjYr for the detailed and constructive comments. In the following, we have provided an item-by-item response to the comments.
>
> > **Q1**: the method introduces extra hyperparameters including weight, reset frequency and lookup updates. It would be beneficial if author and provide heuristics on how to choose those hyperparameters and how sensitive the algorithm are to those parameters.
>
> **A1**: Thanks for your careful reading. We have included extensive ablation studies in **Appendix E** to analyse their effects and sensitity. For your convenience, we briefly summarize the purpose and key conclusions of each section below:
>
> -   **Diffusion Steps**: Appendix E.1 presents the ablation study on the number of diffusion steps $T$. The results indicate that $T = 100$ achieves the best performance in most experiments, making it a reasonable default setting.
>
> -   **Supervised Ratio**: Appendix E.2 provides the ablation study on the supervised ratio $r_s$. The results show that $r_s = 5\\%$, $10\\%$, and $20\\%$ yield comparable performance, whereas $r_s = 0\\%$ and $r_s = 100\\%$ lead to noticeable degradation. Specifically, $r_s = 0\\%$ increases the optimality gap, and $r_s = 100\\%$ causes the method to degenerate into Native Diffusion. Thus, we recommend a default setting of $r_s = 10\\%$.
>
> -   **Training Samples**: Appendix E.3 analyzes the effect of the number of training samples $K_t$. While increasing $K_t$ generally improves performance, it also exhibits diminishing returns. A moderate choice of $K_t = 16$ offers a good trade-off between efficiency and effectiveness.
>
> -   **Evaluation Samples**: Appendix E.4 evaluates the number of evaluation samples $K_e$. Performance improves as $K_e$ increases, though again with diminishing returns. A value of $K_e = 64$ is found to be effective in practice.
>
> -   **Noise Levels**: Appendix E.6 examines the influence of the noise level $\eta$. The results suggest that the effect of $\eta$ interacts with $K_e$; however, following the default DDPM setting, using $\eta = 1$ is generally sufficient.
>
> -   **Reset Mechanism**: Appendix C examines the influence of the reset mechanism. The results suggest that the effect of reset mechanism leads better feasibility. Thus $2$ is a good choice for reset frequency.
>
> We hope this summary clarifies the roles and recommended configurations of these hyperparameters. For detailed ablation experiment, please refer to Appendix.
>
>
> > **Q2**: the author only report evaluation time in the main text. It would be helpful to also report training time for different method to see the overhead introduced by two-stage training.
>
> **A2**: Thank you for your question. The complete training process is presented in Appendix E.5. Here we present the training time of DiOpt on all benchmarks for 10000 epochs.
>
> | Method    | QP (h) | CQP (h) | QPSR (h) | acopf57 (h) | acopf118 (h) | Retargeting (h) |
> | --------- | ------ | ------- | -------- | ----------- | ------------ | --------------- |
> | **DiOpt** | 3.75   | 3.75    | 3.75     | 9.5         | 33           | 1               |
>
> Our experiments were conducted on a workstation equipped with an Intel Core i9-14900K CPU and NVIDIA GeForce RTX 4090 D GPU (each with 24 GB of memory). The system runs Ubuntu 22.04.5 LTS.
>
> > **Q3**: Can DiOPT handle equality constraints?
>
> **A3**: Yes. As shown in line 355-371, we follow a standard technique in learn-to-optimize literature to handle equality constraints. Moreover, the our evaluation benchmarks (including QP, QPSR, CQP, ACOPF57/118) all contain equality constraints.
>
> We also present the conrete technique here for your convenience. We first decompose the variable as $\mathbf{y}=[{\mathbf{y}_b,\mathbf{y}_n}]$. Then, for problems with equality constraints, we just utilize the diffusion to generate partial variables $y_b$ and use an equation solver to solve the left part $y_n$ (the equation is $h_i(\mathbf{y}_b,\mathbf{y}_n;x)=0$). In that case, the equality constraints can be satisfied accurately.

---

> ### Author Response · Authors · 2025-11-23
>
> > **Q4**: How sensitive is DiOPT to parameters like rs and Kt in the paper?
>
> **A4**: As shown in Figure 6 $(r_s)$ and Figure 7 $(K_t)$ of Appendix E, our method is not very sensitive to the value of $r_s$, as long as it is not set to extreme values such as $0\%$ or $100\%$. When $r_s = 2\%, 5\%$, or $20\%$, the performance curves show only minor differences. In contrast, our method is relatively sensitive to $K_t$; if $K_t$ is too small, the performance tends to degrade. However, as $K_t$ increases, the performance improvement gradually diminishes. Therefore, we recommend setting $K_t = 16$, which is sufficient in practice.
>
>
> > **Q5**: In standard constrained optimization, Lagrangian multiplier is employed to control the penalty level based on violation. is it possible to use the same practice to make the weight update for constrains less rely on heuristics?
>
> **A5**: That is a good question. However, in practice, Employing Lagrangian multiplier will introduce extra cost in updates of Lagrangian multipliers. Besides, it also leads the nonstationary target distribution of diffusion due to the change in Lagrangian multipliers. In that case, it introduces additional difficulties in implementation. For this reason, our final strategy is to prioritize feasibility in the weight design, and only after feasibility is ensured do we account for the objective function. This design follows the formulation in Equation (8):
>
> $$
> p(y; x) \sim \mathbb{I}_{C(x)}(y)\exp\bigl(-\beta f(y; x)\bigr). \tag{8}
> $$
>
> Of course, it is truly an interesting idea and we will consider it as part of our future work.

---

> > ### Comment · Reviewer_tjYr · 2025-11-24
> > **Thanks for the additional clearifications**
> >
> > Thank you for the detailed and throughtful responses. I appreciate the clearification on hyperparameter choices. The Langrangian-based constraints update rule could be achieved with more sophisticated design. With that said, I agree with reviewer tdfk that the novelty of weighting method and theorical grounding could be further improved. I will maintain my current score.

---

### Official Review · Reviewer_tDFK · 2025-10-30

**Soundness:** 3
**Presentation:** 3
**Contribution:** 2
**Rating:** 4
**Confidence:** 3

**Summary:**

The paper introduces DiOpt, a training framework enhancing the capability of diffusion models for solving optimization problems. The method introduces sample weighting functions to bias the training towards samples which have lower constraint violation and regret. The evaluation is conducted across several interesting problems, most notably the ACOPF setting, which is a challenging and highly nonconvex real-world application.

**Strengths:**

- **Empirical Analysis:** The evaluation includes several different optimization problems, the majority of which require adherence to nonconvex constraints sets. In particular, I appreciate the results on the ACOPF settings, which are considered to be an important problem; I believe these experiments add real-world significance to the results.

- **Methodological Simplicity:** The method proposed is fairly intuitive, and the analysis motivating its adoption sets this up well. As the results are strong, the simplicity of the approach should be considered a strength of the work.

- **Exposition:** The paper is easy to follow, and positions itself well within the broader literature.

**Weaknesses:**

- **Novelty:** I have some concerns regarding the novelty of the method, considering the similarity to [1]. While the overlap methodologically leads me to view this work as more of an application paper, it is not currently presented this way. I believe this work could differentiate itself better by leaning further into an exploration the specific weighting function used for this domain. From a theoretical perspective, the analysis surrounding the weighting scheme is fairly limited, and, in its current form, this appears to be closer to a heuristic than a grounded rule. Some of the ablations in the appendix do help strengthen the authors' case, and it would be useful if these could be alluded to better in the main paper.

- **Clarifications of Experiments:** The exposition here seems to be the weakest, in part because little space is dedicated to it. Many key details are buried in the appendix. For example, in the appendix it seems that the QP experiment is convex; if this is the case, why does DC3 report such high violations -- I'd expect these to approach zero, based on the description. Can the authors speak to this?

- **Additional Baselines:** It would be interesting to see how the performance compares to methods that integrate hard constraints. Of course, formal guarantees cannot be provided on the nonconvex settings considered, but it would be of interest to provide results on other methods from Table 1.

- **Out-of-Distribution Testing:** Has any analysis been given to OOD settings? I assume that in all settings the model has been trained on the same distribution (e.g., same constraints and objective) as it is tested on. Could conditioning be used to encourage generalization?

---

[1] Ding, Shutong, et al. "Diffusion-based reinforcement learning via q-weighted variational policy optimization." Advances in Neural Information Processing Systems 37 (2024): 53945-53968.

**Questions:**

See questions in weaknesses section.

---

> ### Author Response · Authors · 2025-11-23
>
> Thanks for your valuable comments and suggestions. Here we address your detailed comments as below:
>
> > **Q1(1)**: I have some concerns regarding the novelty of the method, considering the similarity to [R1]. While the overlap methodologically leads me to view this work as more of an application paper, it is not currently presented this way.
>
> **A1(1)**: Thanks for raising the concern. We acknowledge that the weighting mechanism of DiOpt is inspired by QVPO[R1]. However, QVPO and its corresponding weight function are actually designed for online reinforcement learning, while our work focuses on the constrained optimization. They are two irrelevant research domains and the weight function designing are totally different.
>
> Concretely, QVPO derives its weights from the advantage function in reinforcement learning as
>
> $$
> \begin{equation*}
> \omega(s,a) :=
> \begin{cases}
> A(s,a), & A(s,a) \geq 0 \\\\
> 0, & A(s,a) < 0
> \end{cases}
> \end{equation*}
> $$
>
> whereas the weight fucntion of DiOpt originate from
>
> $$
> \begin{equation*}
> \omega(\boldsymbol{y};\boldsymbol{x}) :=
> \begin{cases}
> \exp\left( f^\star(\boldsymbol{x}) - f(\boldsymbol{y};\boldsymbol{x}) \right), & \boldsymbol{y} \in \mathcal{C}(\boldsymbol{x}), \\\\
> -\displaystyle\sum_i \max\left( g_i(\boldsymbol{y};\boldsymbol{x}), 0 \right), & \boldsymbol{y} \notin \mathcal{C}(\boldsymbol{x}).
> \end{cases}
> \end{equation*}
> $$
>
> Therefore, although there are some  similarities in form, DiOpt and QVPO differ substantially in both their application goals and their weight function design.
>
> [R1] Ding, S., Hu, K., Zhang, Z., Ren, K., Zhang, W., Yu, J., ... & Shi, Y. (2024). Diffusion-based reinforcement learning via q-weighted variational policy optimization. Advances in Neural Information Processing Systems, 37, 53945-53968.
>
> > **Q1(2)**: I believe this work could differentiate itself better by leaning further into an exploration the specific weighting function used for this domain. From a theoretical perspective, the analysis surrounding the weighting scheme is fairly limited, and, in its current form, this appears to be closer to a heuristic than a grounded rule. Some of the ablations in the appendix do help strengthen the authors' case, and it would be useful if these could be alluded to better in the main paper.
>
>
> **A1(2)**: Thanks for your insightful comments. In fact, the weighting function is specifically designed for the constrained optimization. The principal idea here is that we hope to deal with the constraint satisfaction first, and then improve the optimality based on that. Hence, according to the weighting function
>
> $$
> \omega(y;x) :=
> \begin{cases}
> \exp\big(f^\star(x) - f(y;x)\big), & y \in \mathcal{C}(x), \\\\
> -\displaystyle\sum_i \max\big(g_i(y;x), 0\big), & y \notin \mathcal{C}(x),
> \end{cases}
> $$
>
> we apply the constraint violation to the infeasible samples and use the energy function of optimality gap as weighting function for feasible samples. In that case, with such a weighting function, the diffusion distribution will finally converge to the target distribution in Equation (8) as
>
> $$
> p(y \mid x) \\propto\\mathbb{I}_{\mathcal{C}(x)}(y) \, \exp\big(-\beta f(y;x)\big).
> $$
>
> Moreover, we also move some of our ablation to the experiment part in main text in the revised version to demonstrate the impact of differennt components (e.g., Reset operation) in DiOpt.
>
> > **Q2(1)**: Clarifications of Experiments: The exposition here seems to be the weakest, in part because little space is dedicated to it. Many key details are buried in the appendix.
>
> **A2(1)**: Thank you for your constructive advice. We have revised our manuscript accordingly and added a concise description of the benchmark in Section 6 (Experiment), including the problem types, constraint structures, and their real-world significance (e.g., ACOPF, Retargeting).

---

> ### Author Response · Authors · 2025-11-23
>
> > **Q2(1)**: Clarifications of Experiments: The exposition here seems to be the weakest, in part because little space is dedicated to it. Many key details are buried in the appendix.
>
> **A2(1)**: Thank you for your constructive advice. We have revised our manuscript accordingly and added a concise description of the benchmark in Section 6 (Experiment), including the problem types, constraint structures, and their real-world significance (e.g., ACOPF, Retargeting).
>
> In addition, we have also added more discussion and analysis of the experimental results. Specifically, we now provide a more detailed analysis of each baseline and the underlying reasons for their performance. We also explicitly illustrated where the corresponding ablation studies can be referred in the appendix.
>
> > **Q2(2)**: For example, in the appendix it seems that the QP experiment is convex; if this is the case, why does DC3 report such high violations -- I'd expect these to approach zero, based on the description. Can the authors speak to this?
>
> **A2(2)**: Thank you for the question. Although the QP in our experiment is convex, **DC3 (a learn-to-optimize method) cannot guarantee fesibility** even for a convex problem. DC3 first predicts an approximate solution and then applies gradient-based corrections to improve feasibility. Note that **our QP tasks** are **more challenging** than those in the DC3 benchmark (e.g., the number of inequality constraints increases from 50 to 250 as stated in Appendix F). This makes it harder for the network to learn predictions that lie sufficiently close to the feasible region for the refinement step to succeed, which contributes to the observed violations.
>
>
> > **Q3(1)**: Additional Baselines: It would be interesting to see how the performance compares to methods that integrate hard constraints.
>
> **A3(1)**: Thank you for the comment. In fact, DC3 [R2] is already proposed as a method designed to handle hard constraints. Therefore, our experiments inherently include a comparison against a method that incorporates hard constraints.
>
> [R2] Donti, P. L., Rolnick, D., & Kolter, J. Z. (2021). DC3: A learning method for optimization **with hard constraints**. arXiv preprint arXiv:2104.12225.
>
>
>
> > **Q3(2)**: Of course, formal guarantees cannot be provided on the nonconvex settings considered, but it would be of interest to provide results on other methods from Table 1.
>
> **A3(2)**: As summarized in Table 1, several representative approaches fall into this category, including DC3, DIFFOPT, MBD, DiffuSolve, RectFlow, and T2T. Among these:
>
> - **DC3 and MBD** are already included as baselines in our experiments.
> - **T2T** is specifically designed for **combinatorial optimization problems**, and therefore does not apply to the **continuous optimization settings** considered in our work.
> - **DIFFOPT** focuses on black-box optimization, which is fundamentally different from our setting and thus not directly comparable.
> - For **DiffuSolve** and **RectFlow**, These two methods are not open-sourced. Besides, both methods rely on supervised learning and share the same modeling paradigm as the Naive Diffusion baseline (supervised diffusion) discussed in Table 5 of Appendix E.7. (Table 6 in the revised version) Consequently, they are expected to exhibit similar limitations, particularly regarding feasibility, and the performance of Naive Diffusion provides a reasonable indication of what one might expect from DiffuSolve and RectFlow in our experimental setup.
>
> We hope this clarifies our baseline selection and the reasons why certain methods from Table 1 were not included in the final comparison. Moreover, we have updated Table 2 in the revised manuscript to incorporate Naive Diffusion as an additional baseline.

---

> > ### Author Response · Authors · 2025-11-23
> >
> > For ease of reference, we also present the updated Table 2 here:
> >
> > | Problem         | Method    | Feasibility (%) ↑ | Gap (%) ↓            | Ineq Max ↓      | Ineq Num Viol ↓ |
> > | --------------- | --------- | ----------------- | -------------------- | --------------- | --------------- |
> > | **QP**          | IPOPT     | 100.00% ± 0.00    | 0.00% ± 0.00         | 0.00 ± 0.00     | 0.00 ± 0.00     |
> > |                 | Diffusion | 0.00% ± 0.00      | 0.97% ± 3.22         | 0.58 ± 0.65     | 14.02 ± 4.20    |
> > |                 | DiOpt     | 79.11% ± 0.00     | 3.45% ± 1.23         | 0.02 ± 0.06     | 0.28 ± 0.50     |
> > |                 | DC3       | 22.93% ± 0.00     | 34.98% ± 6.64        | 0.48 ± 0.64     | 2.09 ± 2.34     |
> > |                 | MBD       | 0.04% ± 0.00      | 3588.43% ± 3743.72   | 81.30 ± 50.46   | 95.14 ± 19.30   |
> > | **QPSR**        | IPOPT     | 100.00% ± 0.00    | 0.00% ± 0.00         | 0.00 ± 0.00     | 0.00 ± 0.00     |
> > |                 | Diffusion | 0.00% ± 0.00      | 0.43% ± 2.87         | 0.34 ± 0.48     | 11.84 ± 2.82    |
> > |                 | DiOpt     | 81.87% ± 0.00     | 2.48% ± 0.78         | 0.01 ± 0.04     | 0.23 ± 0.47     |
> > |                 | DC3       | 20.65% ± 0.00     | 33.61% ± 7.00        | 0.48 ± 0.64     | 2.03 ± 2.18     |
> > |                 | MBD       | 0.04% ± 0.00      | 4101.15% ± 4377.28   | 83.11 ± 53.43   | 95.84 ± 19.66   |
> > | **CQP**         | IPOPT     | 100.00% ± 0.00    | 0.00% ± 0.89         | 0.00 ± 0.00     | 0.00 ± 0.00     |
> > |                 | Diffusion | 0.00% ± 0.00      | 1.47% ± 0.85         | 0.33 ± 0.18     | 11.27 ± 1.90    |
> > |                 | DiOpt     | 69.95% ± 0.00     | 7.04% ± 1.36         | 0.05 ± 0.22     | 0.75 ± 1.82     |
> > |                 | DC3       | 0.00% ± 0.00      | N/A                  | N/A             | N/A             |
> > |                 | MBD       | 0.00% ± 0.00      | 45672.14% ± 23099.09 | 602.93 ± 214.30 | 70.64 ± 5.15    |
> > | **Retargeting** | IPOPT     | 100.00% ± 0.00    | 0.05% ± 1.22         | 0.00 ± 0.00     | 0.00 ± 0.00     |
> > |                 | Diffusion | 100.00% ± 0.00    | 1.24% ± 2.52         | 0.00 ± 0.00     | 0.00 ± 0.00     |
> > |                 | DiOpt     | 100.00% ± 0.00    | 0.65% ± 1.19         | 0.00 ± 0.00     | 0.00 ± 0.00     |
> > |                 | DC3       | 95.86% ± 0.00     | 30.16% ± 37.68       | 0.01 ± 0.06     | 0.04 ± 0.20     |
> > |                 | MBD       | 0.00% ± 0.00      | 169.20% ± 57.09      | 2.30 ± 0.33     | 1.00 ± 0.00     |
> > | **ACOPF57**     | IPOPT     | 100.00% ± 0.00    | 0.00% ± 0.00         | 0.00 ± 0.00     | 0.00 ± 0.00     |
> > |                 | Diffusion | 81.33% ± 0.00     | 0.19% ± 0.71         | 0.01 ± 0.01     | 0.26 ± 0.56     |
> > |                 | DiOpt     | 93.33% ± 0.00     | 0.24% ± 0.91         | 0.00 ± 0.01     | 0.09 ± 0.33     |
> > |                 | DC3       | 94.00% ± 0.00     | 0.40% ± 0.85         | 0.00 ± 0.00     | 0.07 ± 0.29     |
> > |                 | MBD       | 0.00% ± 0.00      | 22.36% ± 14.01       | 1.30 ± 0.54     | 4.30 ± 1.28     |
> > | **ACOPF118**    | IPOPT     | 100.00% ± 0.00    | 0.00% ± 0.00         | 0.00 ± 0.00     | 0.00 ± 0.00     |
> > |                 | Diffusion | 0.00% ± 0.00      | 2.90% ± 0.21         | 0.38 ± 0.14     | 9.10 ± 1.30     |
> > |                 | DiOpt     | 84.33% ± 0.00     | 2.26% ± 0.11         | 0.01 ± 0.01     | 0.20 ± 0.40     |
> > |                 | DC3       | 43.00% ± 0.00     | 2.49% ± 0.19         | 0.02 ± 0.03     | 0.73 ± 0.76     |
> > |                 | MBD       | 0.00% ± 0.00      | 37.39% ± 12.81       | 4.50 ± 2.41     | 23.16 ± 1.33    |

---

> ### Author Response · Authors · 2025-11-23
>
> > **Q4(1)**: Out-of-Distribution Testing: Has any analysis been given to OOD settings? I assume that in all settings the model has been trained on the same distribution (e.g., same constraints and objective) as it is tested on.
>
> **A4(1)**: Thanks for your question. In fact, each optimization problem in our experiments is randomly generated, so there should be no overlap among them. During training, the models are trained exclusively on the **training set**, and all results reported in **Table 2** are evaluated on the **test set**, ensuring that the train and test set do not share the same distribution (e.g., same constraints and objective). The details of our benchmarks can be referred to in **Appendix F**.
>
>
> >**Q4(2)**:Could conditioning be used to encourage generalization?
>
> **A4(2)**: Yes, Our DiOPT does involves conditions for diffusion. As shown in Equation (7), DiOpt is conditioned on the problem parameters $x$. In that case, DiOpt can generate the solution corresponding to the specific problem.

---

### Official Review · Reviewer_2zyr · 2025-10-31

**Soundness:** 3
**Presentation:** 3
**Contribution:** 4
**Rating:** 6
**Confidence:** 2

**Summary:**

The authors discuss solving constrained optimization problems with a generative diffusion model. They show that training the model on a dataset of solutions is not adequate, with the sampled solutions regularly being outside the feasible space. Thus, they propose an alternate training scheme, DiOpt, that mixes "supervised" training with solutions and "unsupervised" training with random points, weighted by their objective value, to steer the diffusion model towards generating samples in the feasible region. The experiments show that the proposed method can be used as a learned solver on a set of constrained optimization tasks.

**Strengths:**

- The idea of training a diffusion model in an "unsupervised" way, i.e., without a ground truth dataset of samples (solutions), is very interesting and has not been explored before to the best of my knowledge. The authors effectively show that with appropriate weighting, they can train the diffusion model to generate samples in the desired region, which in this case, corresponds to the feasible space of solutions to the optimization problem. This could be a significant contribution both to the learned optimization and the overall diffusion crowd.

- DiOpt can effectively produce solutions to the constrained optimization problem, whereas a naive approach of training the diffusion on solutions only fails. Additionally, when compared to two previous baselines, DiOpt consistently achieves results closest to an optimization solver, while only requiring a fraction of the time.

**Weaknesses:**

- The baselines to which the proposed method is compared are not well established in the main text, making it difficult to interpret the results of Table 2. The main result of the paper in Table 2 requires the reader to know how the two baselines (DC3, MBD) work to get a clear picture of the advantages of the proposed algorithm. It seems that DC3 trains a network to perform the optimization, whereas MBD runs some kind of solver within its algorithm (and seems not to work at all). Thus, apart from establishing that DiOpt achieves better results, the experiment does not really provide any information related to what the advantages or critical components of DiOpt are.

- The authors, throughout the paper, compare the proposed DiOpt to the naive approach of training the diffusion model in a supervised way, but do not include results of the "naive diffusion" approach in the main table. Some of those results are found in the appendix ablations. Table 2 should include an additional row with the naive approach to establish a baseline of what the problems of training the diffusion without DiOpt are.

**Questions:**

- You mention six related learned constrained optimization methods from the literature, but end up only comparing to two. Are the other methods not applicable to the settings you have applied DiOpt to, and if not, could you apply DiOpt to their settings?

- What is the training time of DC3 in Table 2, and what is the training time of DiOpt? Additionally, MBD seems to be an inference-only method, meaning that it does not require any training at all. Is the comparison fair in that case?

- Is the issue of the naive diffusion approach the amount of training data? If you had a large enough dataset, would you observe the same issue as shown in Figure 2?

- In Algorithm 1, should line 333 be `if n mod 2 == 0 then`? Does the reset of Equation (11) happen for both infeasible and feasible ($\omega = 0$) points?

---

> ### Author Response · Authors · 2025-11-23
>
> We would like to thank the Reviewer 2zyr for the detailed and constructive comments. In the following, we have provided an item-by-item response to the comments.
>
> > **Q1**: The baselines to which the proposed method is compared are not well established in the main text, making it difficult to interpret the results of Table 2. The main result of the paper in Table 2 requires the reader to know how the two baselines (DC3, MBD) work to get a clear picture of the advantages of the proposed algorithm. It seems that DC3 trains a network to perform the optimization, whereas MBD runs some kind of solver within its algorithm (and seems not to work at all). Thus, apart from establishing that DiOpt achieves better results, the experiment does not really provide any information related to what the advantages or critical components of DiOpt are.
>
> **A1**: Thank you for your insightful comments. Due to page limitation in the main text, we put the detailed descriptions of MBD and DC3 in **Appendix G**. Specifically, DC3 is a learning-based neural solver. It performs *equality completion* to satisfy equality constraints and uses *inequality correction* to push the completed point toward feasibility with respect to inequality constraints. Model-based Diffusion (MBD) is a purely sampling-based method. At each iteration, it samples $d$ candidate points $\mathbf{z}_j$ from a Gaussian distribution and computes a weight $w_j$ for each point. These weights are computed using the same formulation as the DC3 loss above. We have follow your advice and revise the experiemnt section of our paper. You can refer to our new version for more details.
>
> Besides, we have also added Naive Diffusion as a baseline in Table 2 of our revised version to show the superiority of weighted bootstrapping training in DiOpt. It can be observed that, with self-supervised weighted bootstrapping training , DiOpt performs better than supervised diffusion (Naive Diffusion) and other baseline methods in terms of constraint violation and feasiblity rate. For your convenience, we also provide the new Table 2 in **A2**.
>
> > **Q2**: The authors, throughout the paper, compare the proposed DiOpt to the naive approach of training the diffusion model in a supervised way, but do not include results of the "naive diffusion" approach in the main table. Some of those results are found in the appendix ablations. Table 2 should include an additional row with the naive approach to establish a baseline of what the problems of training the diffusion without DiOpt are.
>
> **A2**: Thank you for your valuable suggestion. We have added Naive Diffusion as a baseline in Table 2 of our new version. It can be found that DiOpt has a substantial improvement on Feasibility compared with Naive Diffusion owing to the weighted bootstrapping training.
>
> | Problem     | Method           | Feasibility(%)$\uparrow$ | Gap(%)$\downarrow$ | Time(s)$\downarrow$ |
> | ----------- | ---------------- | ------------------------ | ------------------ | ------------------- |
> | QP          | DiOpt            | **79.11% ± 0.00**        | **3.45% ± 1.23**   | 0.01 ± 0.00         |
> | QP          | Native Diffusion | 0.00% ± 0.00             | 0.97% ± 3.22       | 0.01 ± 0.00         |
> | QPSR        | DiOpt            | **81.87% ± 0.00**        | **2.48% ± 0.78**   | 0.01 ± 0.00         |
> | QPSR        | Native Diffusion | 0.00% ± 0.00             | 0.43% ± 2.87       | 0.01 ± 0.00         |
> | CQP         | DiOpt            | **69.95% ± 0.00**        | **7.04% ± 1.36**   | 0.01 ± 0.00         |
> | CQP         | Native Diffusion | 0.00% ± 0.00             | 1.47% ± 0.85       | 0.01 ± 0.00         |
> | RETARGETING | DiOpt            | **100.00% ± 0.00**       | **0.65% ± 1.19**   | 0.01 ± 0.00         |
> | RETARGETING | Native Diffusion | 100.00% ± 0.00           | 1.24% ± 2.52       | 0.01 ± 0.00         |
> | ACOPF57     | DiOpt            | 93.33% ± 0.00            | 0.24% ± 0.91       | 0.02 ± 0.00         |
> | ACOPF57     | Native Diffusion | 81.33% ± 0.00            | **0.19% ± 0.71**   | 0.02 ± 0.00         |
> | ACOPF118    | DiOpt            | **84.33% ± 0.00**        | **2.26% ± 0.11**   | 0.07 ± 0.00         |
> | ACOPF118    | Native Diffusion | 0.00% ± 0.00             | 2.90% ± 0.21       | 0.07 ± 0.00         |

---

> ### Author Response · Authors · 2025-11-23
>
> > **Q3**: You mention six related learned constrained optimization methods from the literature, but end up only comparing to two. Are the other methods not applicable to the settings you have applied DiOpt to, and if not, could you apply DiOpt to their settings?
>
> **A3**: Thanks for raising the concern. As we mentioned in line 143, **T2T** is proposed to utilize diffusion to solve combintorial optimization problems, which is totally not applicable to our continuous constrained optimization setting. As to **DiffOpt**, **DiffuSolve** and **RectFlow**, we do not include them in the experiments since their official implementation are unavailable. However, their performance should be similar to the Naive Diffusion in constrained continuous optimization tasks since they are all implemented in a supervised paradigm.
>
>
> > **Q4(1)**: What is the training time of DC3 in Table 2, and what is the training time of DiOpt?
>
> **A4(1)**: The training times for DiOpt and DC3 across various tasks are reported in the table below. All models were uniformly trained for 10,000 epochs.
>
> DiOpt requires additional training time compared to DC3 under the same training iteration since diffusion itself needs to peform an iterative denoising procedure in the inference phase. However, DiOpt has a low inference time close to DC3 as shown in Table 2.
> | Method       | QP       | CQP      | QPSR     | acopf57 | acopf118 | Retargeting |
> |--------------|----------|----------|----------|---------|----------|-------------|
> | DC3          | 0.67h    | 0.67h    | 0.67h    | 2.52h   | 13.05h   | 0.33h       |
> | DiOpt        | 3.75h    | 3.75h    | 3.75h    | 9.48h   | 33.12h   | 1.02h       |
>
>
>
> > **Q4(2)**: Additionally, MBD seems to be an inference-only method, meaning that it does not require any training at all. Is the comparison fair in that case?
>
> **A4(2)**: Thank you for your valuable feedback. MBD is a milestone in the leaning-to-optimize field that incorporates diffusion into the soft-constrained optimization problems. In that case, although MBD is an inference-only method, it is necessary to use MBD as a baseline algorithm.
>
> Besides, according to Table 2, MBD usually cost much more time during the inference. This is because it need to access the objective and constraints at each denosing step of diffusion. It is somewhat like a traditional solver e.g., IPOPT, which do not require training but have a high inference cost. From this perspective, it makes sense to add MBD as a baseline for comparison.
>
>
> > **Q5**: Is the issue of the naive diffusion approach the amount of training data? If you had a large enough dataset, would you observe the same issue as shown in Figure 2?
>
> **A5**: No, this issue is not related to the amount of training data, but the inherent limitation of the learning-based model, which introduces the approximation error that follows a Gaussian distribution. This results in the generated points being around the optimum rather than within the feasible region. This further leads to its low feasibility. Thus, even if we have a large enough dataset, the same issue can be still observed as shown in Figure 2 as well. Our theoretical analysis in Appendix A and Appendix B can also confirm that.
>
> > **Q6(1)**: In Algorithm 1, should line 333 be `if n mod 2 == 0` then?
>
> **A6(1)**: Thanks for your feedback. It is a typo. We have revised it in the new vesion.
>
> > **Q6(2)**: Does the reset of Equation (11) happen for both infeasible and feasible ($\omega=0$) points?
>
> **A6(2)**: Yes, the reset operation happen for both infeasible and feasible points. This mechanism is to prevent DiOpt from degenerating back into vanilla diffusion. The corresponding theoretical analysis and ablation studies are provided in Appendix C. We also moved some of this content to the experiment section for better presentation. For convenience, the results from Table 3 are reproduced below:
>
> | Method | Reset=False          | Reset=True           |
> |--------|----------------------|----------------------|
> | QP     | 75.39% ± 0.00        | 86.63% ± 0.00        |
> | QPSR   | 70.03% ± 0.00        | 81.19% ± 0.00        |
> | CQP    | 83.79% ± 0.00        | 87.07% ± 0.00        |

---

> > ### Comment · Reviewer_2zyr · 2025-11-26
> >
> > Thank you for your very detailed responses.
> >
> > The concerns I raised in my review have been addressed, thus I would be willing to increase my score in my final review.

---

### Author Response · Authors · 2025-11-24

We thank all reviewers for their careful reading and constructive comments. We have updated a revised version of our paper and present the official comments for each of your questions.

---

### Author Response · Authors · 2025-12-03
**A Brief Summary of Rebuttal**

Dear AC,

We sincerely appreciate the time and effort you have devoted to reviewing our paper. For your reference, we provide a brief summary of the reviewers’ comments and concerns during the discussion phase, along with how we addressed each of them.

During the discussion phase, both Reviewer 2zyr and tjYr acknowledge that we have addressed their concerns. Specifically, **Reviewer 2zyr expressed a willingness to raise his/her score from 6**, and **Reviewer tjYr maintained a positive rating of 6**.

The remaining two reviewers (**tDFK**, score: 4; and **iA95**, score: 2) did not get a chance to respond to our rebuttal before the system was closed. Concretely, the negative score 2 from Reviewer iA95 appear to be driven by misunderstandings of DiOpt (**as a general diffusion-based solver for nonconvex problems**), such as the impression that DiOpt is limited to convex problems; and the negative score 4 from Reviewer tDFK mainly results from the concern regarding the **similarity of our work to an online reinforcement learning method**. To address these misunderstandings and concerns, we provide point-by-point clarifications, and we believe our detailed responses and updated PDF file have addressed all of them.

Here, we summarized the main concerns, our corresponding responses, and the views of the other reviewers on these points:


1. **Applicability to Non-Convex Problems (Reviewer iA95)**: The reviewer incorrectly assumed DiOpt is limited to linear/convex settings. We clarified that **most of our benchmarks are nonconvex (e.g., QPSR, Retargeting, and ACOPF)**, and DiOpt consistently outperforms other baseline algorithms in these benchmarks. This choice of challenging benchmarks was also specifically praised by Reviewer tDFK, who noted the ACOPF setting is "**a challenging and highly nonconvex real-world application**" that adds "significance to the results."

2. **Hard-constraint Satisfaction Mechanism (Reviewers iA95, tjYr)**: We clarified that we apply different approaches to handling equality and inequality constraints: **Equality constraints can be satisfied exactly via the variable decomposition technique, while inequality constraints are handled via our novel weighted bootstrapping.** Besides, we also add the 'Naive Diffusion' (supervised diffusion) baseline into the main text to demonstrate that the weighted bootstrapping training of DiOpt can resolve the inherent feasibility limitations of prior diffusion-based methods for optimization. The empirical results illustrate the superiority of DiOpt, which **increases the feasibility from 0% to 82% on the challenging ACOPF118 task**. Moreover, Reviewer 2zyr and tjYr both acknowledge that DiOpt "**can effectively produce feasible solutions**".

3. **Novelty of Weighting Mechanism (Reviewers tDFK, iA95)**: Regarding the concern about similarity to QVPO (**an online RL method**), we clarified that DiOpt operates in a fundamentally different domain (**Constrained Optimization**) with a distinct weight function design and extra tricks such as reset operation and look-up table. Unlike RL advantage functions, our weight function is specifically **tailored to prioritize feasibility before optimizing the objective, addressing the "feasibility collapse" issue**. The novelty of DiOpt is also supported by the comments of Reviewer 2zyr, which states that training a diffusion model in this unsupervised way "is very interesting and has **not been explored before**" and constitutes "a significant contribution."

4. **Theoretical Rigorousness (Reviewer iA95)**: We have clarified that **complete and rigorous proofs analyzing the feasibility collapse in supervised diffusion were already provided in the Appendix and explicitly referenced in the main text.** Furthermore, our empirical results strongly support these theoretical conclusions. This alignment between theory and practice is also recognized by Reviewer tjYr, who noted that "the proposed method is empirically validated on diverse constrained optimization problems with detailed ablation."

5. **Out-of-Distribution & Generalization (Reviewer tDFK)**: We addressed this concern regarding OOD settings with the clarification on our experimental settings and the data resources. As detailed in the Appendix, each problem instance is randomly generated, **ensuring strict separation between training and testing sets**. Consequently, our results can demonstrate the ability of DiOpt to **solve unseen instances with distinct constraints and objectives, rather than memorizing training data**. We further emphasized that this generalization is enabled by our architecture design, where DiOpt is explicitly conditioned on problem parameters $\mathbf{x}$, allowing it to adaptively generate solutions for new problem specifications.

Finally, we would like to sincerely thank you for the time and effort on our work again. More details related to the above illustration can be found in our rebuttals to each reviewer.

The Authors

---

### Meta-Review · Area_Chair_qstk · 2026-01-04

**Summary:**

This paper introduces DiOpt, a training framework that enhances the capabilities of diffusion models for solving constrained optimization problems. Four reviewers give quite divergent reviews. Both Reviewers 2zyr and tjYr acknowledge the contributions of the paper, while two reviewers give negative scores.

Both Reviewers tDFK and iA95 concern the novelty. Specifically, Reviewer tDFK has concerns about the relationship to an online reinforcement learning method, which leads to the fact that the analysis surrounding the weighting scheme is fairly limited. The authors give detailed replies to this concern, which I think, might somehow relieve the concern about novelty. However, the major concerns raised by Reviewers iA95 seem not easy to address in a short time and in the current revisions. I agree with the author's rebuttal that Reviewers iA95 did not find that DiOpt is able to solve non-convex problems, but more importantly, I do agree with Reviewers iA95 that "a contribution of this paper is rather underwhelming" and "The paper could also benefit from more high-level, intuitive explanations of the core scientific idea." In summary, I think this paper is below the borderline of ICLR.

**Reviewer Concerns:**

Most concerns are well addressed by the authors, and they provide detailed responses to these rebuttals.

**Reviewer Scores:**

I appreciate the authors' efforts to address the concerns raised in the rebuttal, and I think Reviewer iA95 might increase their score.

---

### Decision · Program_Chairs · 2026-01-26

Reject